# KSHV infection of endothelial precursor cells with lymphatic characteristics as a novel model for translational Kaposi's sarcoma studies

Krista Tuohinto[1⊛], Terri A. DiMaio[2⊛], Elina A. Kiss[1], Pirjo Laakkonen[1,3], Pipsa Saharinen[1,4], Tara Karnezis[5], Michael Lagunoff[2]*, Päivi M. Ojala[1]*

**1** Translational Cancer Medicine Research Program, Faculty of Medicine, University of Helsinki, Helsinki, Finland, **2** Department of Microbiology, University of Washington, Seattle, WA, United States of America, **3** Laboratory Animal Center, HiLIFE, University of Helsinki, Helsinki, Finland, **4** Wihuri Research Institute, Biomedicum Helsinki, Helsinki, Finland, **5** Gertrude Biomedical Pty Ltd., Melbourne, Victoria, Australia

⊛ These authors contributed equally to this work.
* lagunoff@uw.edu (ML); paivi.ojala@helsinki.fi (PMO)

**Data Availability Statement:** All raw and processed files from expression profiling by high throughput sequencing are available from the

## Abstract

Kaposi's sarcoma herpesvirus (KSHV) is the etiologic agent of Kaposi's sarcoma (KS), a hyperplasia consisting of enlarged malformed vasculature and spindle-shaped cells, the main proliferative component of KS. While spindle cells express markers of lymphatic and blood endothelium, the origin of spindle cells is unknown. Endothelial precursor cells have been proposed as the source of spindle cells. We previously identified two types of circulating endothelial colony forming cells (ECFCs), ones that expressed markers of blood endothelium and ones that expressed markers of lymphatic endothelium. Here we examined both blood and lymphatic ECFCs infected with KSHV. Lymphatic ECFCs are significantly more susceptible to KSHV infection than the blood ECFCs and maintain the viral episomes during passage in culture while the blood ECFCs lose the viral episome. Only the KSHV-infected lymphatic ECFCs (K-ECFCLY) grew to small multicellular colonies in soft agar whereas the infected blood ECFCs and all uninfected ECFCs failed to proliferate. The K-ECFCLYs express high levels of SOX18, which supported the maintenance of high copy number of KSHV genomes. When implanted subcutaneously into NSG mice, the K-ECFCLYs persisted in vivo and recapitulated the phenotype of KS tumor cells with high number of viral genome copies and spindling morphology. These spindle cell hallmarks were significantly reduced when mice were treated with SOX18 inhibitor, SM4. These data suggest that KSHV-infected lymphatic ECFCs can be utilized as a KSHV infection model for in vivo translational studies to test novel inhibitors representing potential treatment modalities for KS.

NCBI's GEO database (accession numbers GSE54416, GSE207589 and GSE207657).

**Funding:** K.T was supported by the Doctoral Program in Biomedicine, University of Helsinki. T. A.D received no specific funding for this work. E.A. K was funded by Academy of Finland (308663). P.L and P.S received no specific funding for this work. T.K is employed by The Gertrude Biomedical Pty Ltd. M.L. was funded by grants (RO1CA240479 and RO1097934) from the National Cancer Institute of the NIH, and P.M.O obtained support from the Gertrude Biomedical Pty Ltd and the Finnish Cancer Foundation. Gertrude Biomedical Pty Ltd. participated in the study design, analysis, decision to publish, and preparation of the manuscript. The other funders had no role in study design data collection and analysis, decision to publish, or preparation of the manuscript.

**Competing interests:** I have read the journal's policy and the authors of this manuscript have the following competing interests: Gertrude Biomedical Pty Ltd. participated in the study design, analysis, decision to publish, and preparation of the manuscript.

## Author summary

Kaposi's sarcoma herpesvirus (KSHV) is the etiologic agent of Kaposi's sarcoma (KS). The main proliferative component of KS, spindle cells, express markers of lymphatic and blood endothelium. Endothelial precursor cells, which are circulating endothelial colony forming cells (ECFCs), have been proposed as the source of spindle cells. Here we examined both blood and lymphatic ECFCs infected with KSHV. Lymphatic ECFCs are readily infected by KSHV, maintain the viral episomes and show modest transformation of the cells, which the infected blood ECFCs and all uninfected ECFCs failed to show. The lymphatic ECFCs express SOX18, which supported the maintenance of high copy numbers of KSHV genomes. The KSHV-infected lymphatic ECFCs persisted in vivo and recapitulated the phenotype of KS tumor cells such as high number of viral genome copies and spindling morphology. These KS tumor cell hallmarks were significantly reduced by SOX18 chemical inhibition using a small molecule SM4 treatment. These data suggest that KSHV-infected lymphatic ECFCs could be the progenitors of KS spindle cells and are a promising model for the translational studies to develop new therapies for KS.

## Introduction

Kaposi's sarcoma herpesvirus (KSHV) is a gamma herpesvirus and the etiologic agent of Kaposi's sarcoma (KS) as well as rare B-cell proliferative diseases primary effusion lymphoma and AIDS-associated Castleman's disease. KS is a hyperplasia consisting of enlarged malformed vasculature and spindle-shaped cells which are the main proliferative component of KS. Spindle cells express markers consistent with an endothelial cell origin [1–5].

Endothelial cells are largely divided into blood vascular and lymphatic endothelial cells. While closely related, the cells making up the blood and lymphatic vasculatures have distinct functions and express specific markers. KS spindle cells express several markers specific to lymphatic endothelium though they also express markers of blood endothelium. However, it has been shown that KSHV infection of endothelial cells in culture induces changes in endothelial cell marker expression. Specifically, KSHV infection of blood vascular endothelial cells (BEC) induces changes consistent with lymphatic differentiation including induction of PROX1 and vascular endothelial growth factor 3, VEGFR3 expression [6–8]. However, infection of lymphatic endothelial cells (LEC) induces the expression of VEGFR1, consistent with its expression in BEC. Therefore, the contribution of these cell types to development of KS tumors is unclear. Another feature of spindle cells is the expression of mesenchymal cell markers, an observation that has prompted the idea that undifferentiated mesenchymal cells might be a cell type of origin [9]. This hypothesis remains elusive since the mesenchymal phenotypes could also emerge due to Endothelial-to-Mesenchymal-Transition (EndMT) induced by KSHV-infection of LECs [10,11].

KS tumors develop primarily in the skin of patients with classic KS. More aggressive AIDS-associated KS can infiltrate internal organs. However, KS is not found in tissues lacking lymphatic vasculature, such as parenchyma and retina. This suggests that LECs are a necessary component of KS development. Furthermore, we and others have previously found that LECs are more susceptible than BECs to KSHV infection [8,12–15]. The KSHV-infected primary human LECs (K-LECs) display a unique infection program characterized by maintenance of high KSHV genome copies, spontaneous lytic gene expression and release of significant amounts of infectious virus [12,13]. This is in striking contrast to BECs and other KSHV-infected in vitro cell models. KSHV infection of neonatal LECs but not BECs allows them to

bypass replicative senescence [14]. However, KSHV-infected LECs, like KS spindle cells, are not fully immortalized. Whether LECs are a reservoir for KSHV *in vivo* has yet to be determined.

Endothelial colony forming cells (ECFCs) are circulating cells thought to be endothelial precursors. Previous studies have shown that circulating endothelial precursor cells home to sites of neoangiogenesis [16,17]. However, the contribution of precursor cells to neoangiogenesis is unclear. Previous investigations suggest that circulating endothelial precursors could be important for the development of KS lesions. One study found that KS lesion formation following kidney transplant occurred in distal extremities [18]. Importantly, the spindle cells of the KS tumors were gender matched to the transplanted tissue, not the recipient, suggesting the presence of circulating cells harboring KSHV. Further studies have isolated ECFCs from patients with classic KS, which were found to be positive for KSHV DNA [19]. Interestingly, KSHV infection of endothelial progenitor cells isolated from umbilical cord blood reduces their angiogenic potential [20]. It has been previously proposed that KSHV infected ECFCs could be the source of spindle cells seeding the KS tumors [19–21].

We previously isolated ECFCs from whole blood and found that there are two types of ECFCs [22]. The predominant ECFC expressed markers of blood endothelial cells as has been previously described [23–25]. Importantly, we identified rare ECFC isolates expressing markers of the lymphatic vasculature. Our recent results show that the key developmental lymphatic transcription factor (TF), SOX18, is expressed in KS tumors and needed to support a unique KSHV infection program with high number of intracellular KSHV genome copies in infected human LECs. SOX18 binds to the origins of KSHV replication and increases the intracellular viral DNA genome copies [12]. Depletion by RNAi or specific inhibition of SOX18 homo- and heterodimerization by a small molecule inhibitor SM4 or R-propranolol [26,27] dramatically decreases both intracellular viral genome copies and release of infectious virus, suggesting SOX18 as an attractive therapeutic target for KS.

Here, we show that similar to LECs, lymphatic ECFCs are more susceptible to KSHV infection and maintain the viral episome in contrast to blood ECFCs. KSHV infection of lymphatic ECFCs allowed proliferation, albeit limited, of these cells in soft agar, suggesting that KSHV may promote enhanced survival of lymphatic ECFCs. We further utilized the SOX18-expressing lymphatic ECFCs as a physiologically relevant KSHV infection model for testing the translational potential of SOX18 small molecule inhibitor SM4 in vitro and in vivo. Together, these data suggest that circulating lymphatic ECFCs could potentially represent virus reservoirs and putative precursors as the source for the initiation of KS tumors.

## Results

### Lymphatic ECFCs are more permissive to KSHV infection than blood ECFCs

We have previously shown that neonatal LECs are more susceptible to KSHV infection than BECs [12,14]. Also, we previously identified two types of ECFCs, ones that expressed markers of BEC and ones that expressed markers of LEC. To determine whether these lymphatic ECFCs (ECFCLYs) resemble neonatal LECs with respect to an enhanced susceptibility to KSHV-infection, we seeded both ECFCLYs and blood ECFCs (ECFCBLs) onto 6-well plates, allowed them to adhere, and followed by infection with identical dilutions of KSHV at either a high or low MOI. After two days, we harvested the cells and performed immunofluorescence for viral latency-associated nuclear antigen (LANA) expression and counted the percentage of cells that were latently infected. As shown in Fig 1A, at high MOI, 98% of ECFCLYs are infected, while only 87% of the ECFCBLs are infected. At high MOI, each cell may have

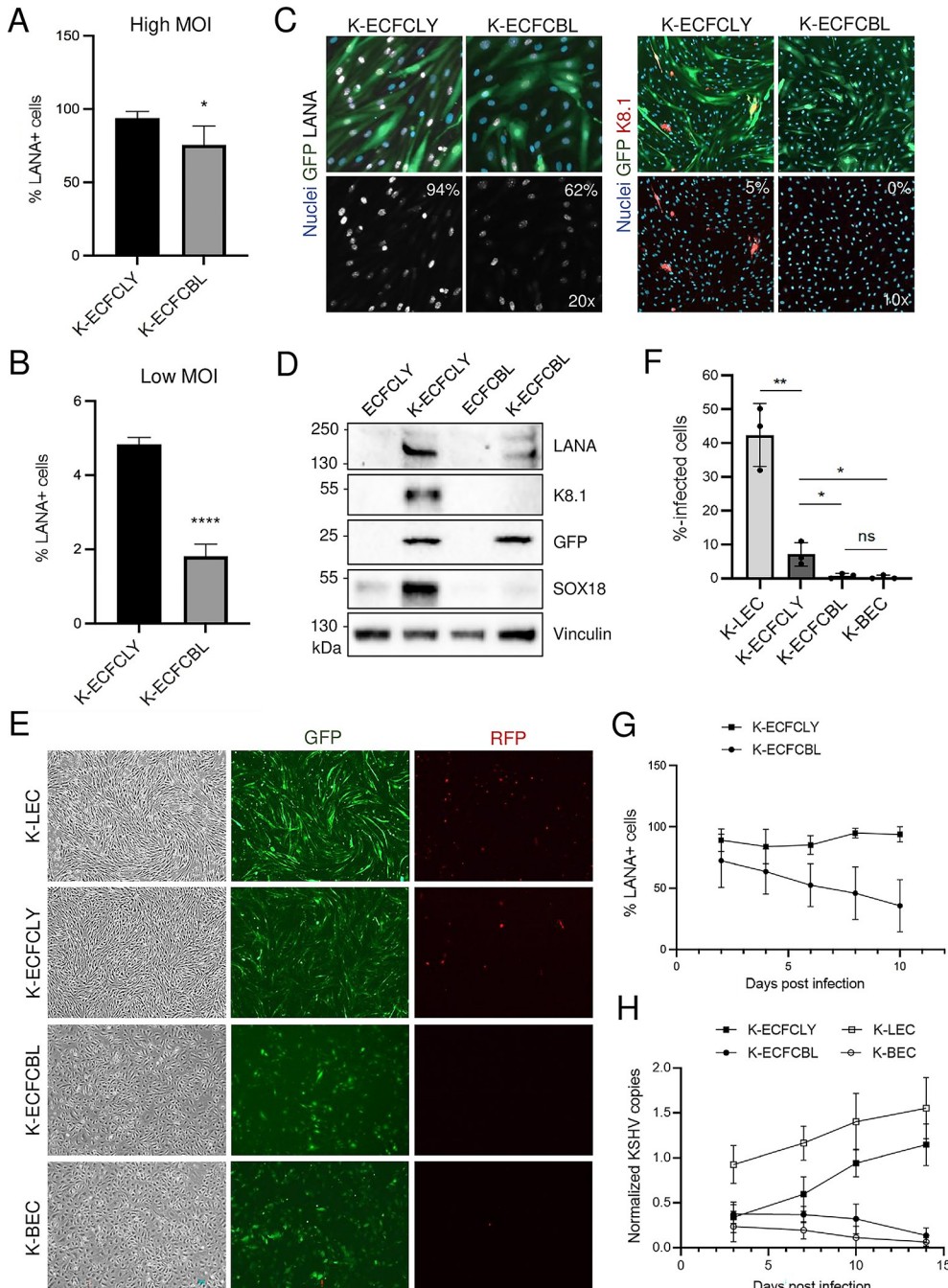

**Fig 1. Lymphatic ECFCs are more permissive to KSHV infection than blood ECFCs and maintain the KSHV viral episome. A.** Lymphatic and blood ECFCs were infected with wtKSHV (K-ECFCLY and K-ECFCBL respectively) at either a high MOI (upper panel) or **B.** a low MOI (lower panel). At 48 h.p.i cells were harvested and stained with anti-LANA antibody for the number of infected cells and DAPI for the number of total cells. Cells were counted and the percentage of LANA+ cells was determined. These experiments were performed with cells isolated from at least two different donors for each cell type. **C.** IF images of rKSHV.219-infected lymphatic and blood ECFCs analysed for the percentage of cells expressing KSHV latent (LANA) and lytic (K8.1) proteins at 5 d.p.i. **D.** Immunoblot of KSHV LANA and K8.1 proteins, GFP and SOX18 from mock- and KSHV-infected (7 d.p.i) lymphatic and blood ECFCs. Vinculin was used as a loading control. **E.** Images of rKSHV.219-infected lymphatic and blood ECFCs, LECs and BECs at 7 d.p.i and showing the spindling phenotype (left panels), latent infection (GFP; middle panels) and spontaneous lytic replication (RFP expression; right panels). **F.** KSHV titers determined by a virus release assay on naïve U2OS cells. **G.** Lymphatic (squares) and blood (circles) ECFCs were infected with wtKSHV at separate high MOI necessary to achieve similar initial infection rates and harvested every two days and stained for LANA expression. The percentage

of LANA+ cells was determined by fluorescence microscopy. **H.** Lymphatic ECFCs (black squares), blood ECFCs (black circles), K-LECs (white squares) and K-BECs (white circles) were infected with rKSHV.219 and total DNA was collected at 3, 7, 10 and 14 d.p.i. Relative KSHV genome copies were determined. Experiments were performed three times with similar results. * $p < 0.05$, ** $p < 0.01$, **** $p < 0.0001$.

multiple instances of infection, therefore counting the percentage of LANA+ cells may miss cases of superinfection. To address this, we performed the same experiment with a low MOI. Fig 1B shows that ECFCLYs are infected at almost three times the rate of ECFCBLs.

We also isolated ECFCs from the whole blood of four, additional healthy donors, but as adherent subpopulations rather than from single cell colonies. By fluorescence associated cell sorting (FACS) analyses these cells expressed substantial levels of CD34, VEGFR3, podoplanin, CD31/PECAM1, PROX1 and SOX18, suggesting that they also represent predominantly lymphatic ECFCs (S1 Fig), not blood vascular endothelial ECFCs. Next, we analyzed if either population of ECFCs could support spontaneous lytic replication and production of new infectious viruses, a phenotype typically observed only in KSHV-infected LECs, but not in BECs [12,13]. To this end, we infected isolated ECFCLYs and ECFCBLs using a low MOI of recombinant KSHV.219 (rKSHV.219) [28] on a 96-well plate. The cells were labeled with antibodies against latent LANA and viral lytic protein K8.1 at five days post infection. As shown in Fig 1C, the infected ECFCLYs had a higher percentage of LANA-positive, latently infected cells, and spontaneously lytic cells expressing K8.1, in contrast to the strictly latent K-ECFCBLs. Accordingly, as shown by immunoblotting in Fig 1D, K-ECFCLYs expressed GFP (indicating latent KSHV infection), LANA, and K8.1, whereas K-ECFCBLs did not. Importantly, KSHV infection led to a clear increase in SOX18 protein levels in ECFCLYs, similar to what we have previously reported in LECs [12]. To further compare the infection program and phenotype of the ECFCLYs and ECFCBLs to mature primary human LECs and BECs, we infected all cell types with rKSHV.219 on 6-well plates and the infection was followed for two weeks. Fig 1E shows at seven days post infection latent infection (indicated by GFP expression) in all cell types, however, the distinct spindling phenotype was seen only in the K-ECFCLYs and K-LECs, but not in K-ECFCBLs or K-BECs. Spontaneous lytic replication (indicated by RFP expression) was observed both in K-ECFCLYs and K-LECs, albeit more prominent in K-LECs, but not seen in either K-ECFCBLs or K-BECs. Moreover, similar to K-BECs, K-ECFCBLs did not produce infectious virus, whereas K-ECFCLYs did albeit significantly less than K-LECs (Fig 1F). Comparable results showing that K-ECFCLYs can spontaneously produce infectious virus were obtained when using ECFCLYs from three additional donors (S2 Fig). This indicates that the lymphatic ECFCs are permissive to KSHV infection and capable of supporting productive, lytic replication.

## Lymphatic ECFCs maintain the KSHV viral episome

The KSHV genome is maintained as an episome in dividing cells, though this process is not robust, and the episome is lost over time in cultured endothelial cells [29–32]. Our previous data with neonatal ECs showed that LECs were able to maintain the KSHV episome but BECs lost the genome relatively rapidly over the course of cell passaging [12,14]. To test whether the rate of loss is different between blood and lymphatic ECFCs, we infected cells with KSHV and harvested them every two days for immunofluorescence for LANA expression. Because lymphatic ECFCs are more susceptible to KSHV infection, we used higher MOI to the blood ECFCs to achieve similar initial infection rates. By using LANA staining as a marker for infection, Fig 1G shows that, although in the beginning very similar infection rates were observed for both blood and lymphatic ECFCs, the blood ECFCs gradually lose the episomes and by ten

days post infection fewer than 40% of cells exhibited punctate LANA staining. In contrast, the lymphatic ECFCs maintained infection at almost 100%.

Additionally, primary LECs and BECs and the lymphatic and blood ECFCs were infected with rKSHV.219 using low MOI of 1–2 and harvested for total DNA isolation at days 3, 7, 10 and 14 post infection. Quantification of KSHV genome copies demonstrated that lymphatic ECFCs, similar to LECs, maintain the viral episomes (Fig 1H). Interestingly, K-ECFCLYs and K-LECs showed even an increase in genome copies by day 14. In contrast, by using low MOI, K-BECs and K-ECFCBLs harbored a lower number of genome copies which were gradually lost during the two weeks of infection (Fig 1H).

## KSHV inhibits proliferation but not tube formation of ECFCs

To determine if KSHV confers a growth advantage to ECFCs, we mock-or KSHV-infected blood and lymphatic ECFCs and seeded $3x10^4$ cells 48 hours post infection in 6-well plates and monitored their confluence over time using the live-cell Essen Bioscience IncuCyte imaging system. Fig 2A shows that mock-infected blood ECFCs proliferated at a slightly higher rate than mock-infected lymphatic ECFCs. Interestingly, KSHV infection significantly reduced proliferation of both blood and lymphatic ECFCs. However, there are no significant differences in the reduction of proliferation between these two cell types suggesting that KSHV has a mild inhibitory effect on cell proliferation, but this is not dependent on blood versus lymphatic cell origin. We then also compared the proliferation rates of lymphatic ECFCs and mature LECs +/- KSHV-infection by seeding cells to 96-well plates for five days and subjecting them to an EdU pulse for 4h before fixation. The Click-IT EdU staining, measured only from LANA+ cells, indicated a slight, but not significant decrease in proliferating cells between the uninfected and KSHV-infected lymphatic ECFCs or between the uninfected and KSHV-infected LECs (Fig 2B), confirming the mild inhibitory effect of KSHV-infection on endothelial cell proliferation. However, the lymphatic ECFCs had an overall significantly higher proliferation rate than the mature LECs.

To determine if KSHV-infection would support viability of lymphatic and blood ECFCs under limited growth factor conditions, mock and KSHV-infected cells were seeded on 96-well plates. The cells were grown either in normal media containing all the supplements, growth factor reduced media or media without any supplements or serum for a 7-day period (media replenished on day four). As shown in Fig 2C, KSHV-infected ECFCLYs survived better than the mock cells or K-ECFCBLs under both the growth factor reduced and serum deprived conditions, indicating increased viability in limiting growth media. We further compared the viability of K-ECFCLYs to similarly infected and conditioned K-LECs at day seven. As shown in Fig 2D, K-ECFCLYs survived significantly better than the K-LECs, which are, however, more lytic in nature.

KS tumors are highly vascularized with irregular, enlarged vascular slits, indicating increased neovascularization. We have previously shown that KSHV infection of ECs promotes the stability of capillary networks in a Matrigel model of endothelial cell migration [33]. To measure whether KSHV alters the ability of blood or lymphatic ECFCs to undergo angiogenesis, we mock- or KSHV-infected blood and lymphatic ECFCs. Cells were plated on a Matrigel matrix 48 hours post infection. Fig 2E and 2F show that both blood and lymphatic ECFCs organized into networks on Matrigel at 4 hours post plating (top panels). Interestingly, unlike mature endothelial cells, both blood and lymphatic ECFCs maintained the cord structures at 24 hours post plating (bottom panels). Under these conditions, KSHV had little effect (Fig 2F) on the ability of ECFCs to organize on Matrigel and maintain their capillary stability.

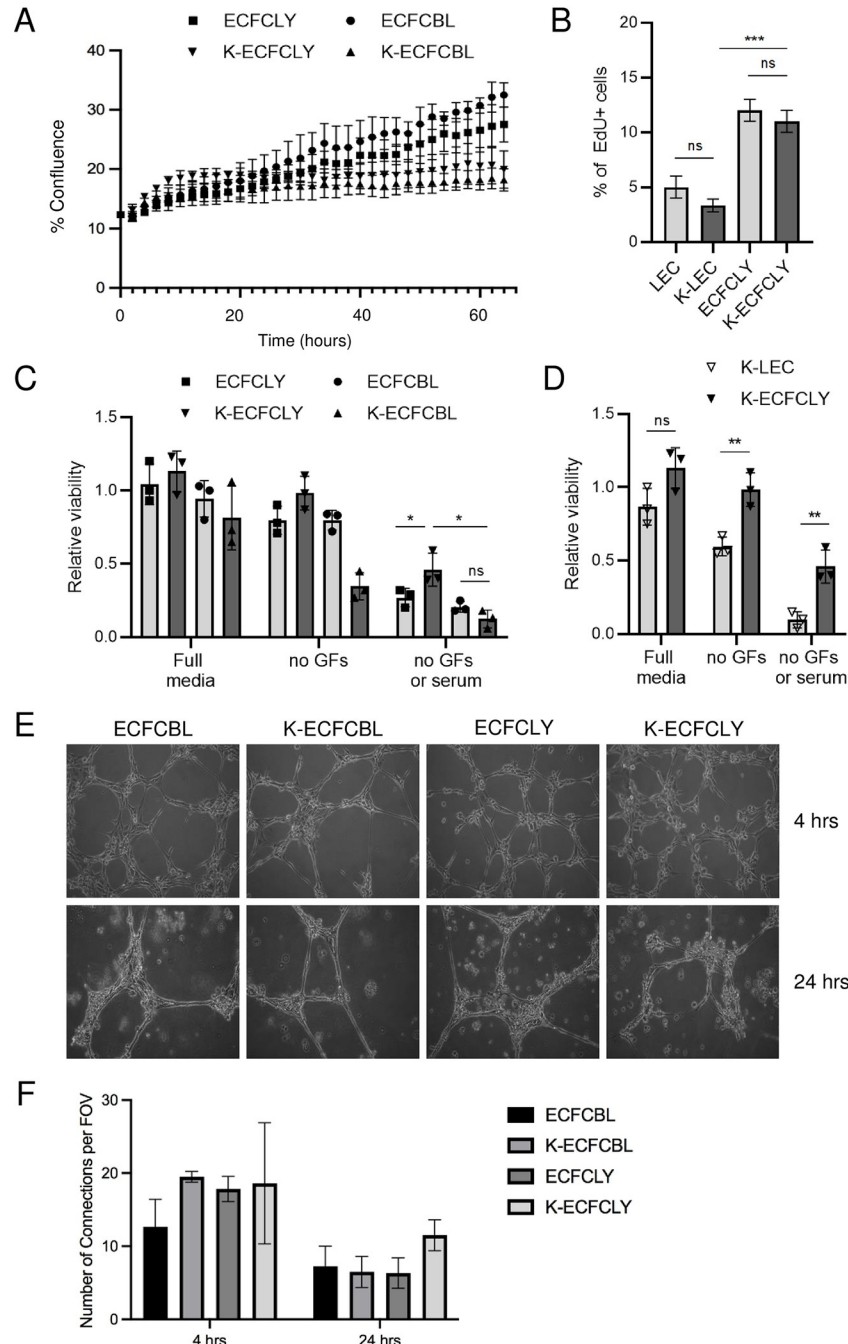

**Fig 2. KSHV inhibits proliferation but not capillary morphogenesis. A.** 48 hours after infection with KSHV, ECFCLY (squares), ECFCBL (circles), K-ECFCLY (upside down triangles) and K-ECFCBL (triangles), were seeded in 6-well dishes and placed in an Essen Biosciences IncuCyte. Pictures were taken every two hours and the percentage of confluence was determined. **B.** Mock- or KSHV-infected ECFCLYs or LECs were seeded 5 d.p.i on 96-well dishes and treated with EdU for 4h before fixing. EdU-positive cells were stained, and percentages of proliferating cells were determined. **C.** Indicated cell types mock or KSHV-infected as in B were grown for seven days in full or serum and/or growth factor deprived media and measured for relative viability by a CellTiter-Glo luminescence assay. D. Relative viability of K-LECs and K-ECFCLYs grown and analyzed as in C. **E.** Blood and lymphatic ECFCs were mock- or wtKSHV-infected. 48 h.p.i cells were plated on Matrigel and monitored for capillary morphogenesis at 4 and 24 hours post plating and **F.** number of branches per field was quantified. * $p < 0.05$, ** $p < 0.01$, *** $p < 0.001$.

## KSHV confers a survival advantage to lymphatic ECFCs grown in soft agar

We next examined if KSHV promotes anchorage independent growth of ECFCs in soft agar, an indicator of cell transformation. We mock- or KSHV-infected blood and lymphatic ECFCs and, after 48 hours, embedded a single cell suspension in a soft agar matrix which was overlaid by normal growth media. We then monitored the growth and formation of small colonies of cells over the course of four weeks. Fig 3A shows images of cell suspensions following four to five weeks of growth. We could not find any evidence of multicellular colonies in the uninfected blood nor lymphatic ECFCs. Only single cells could be visualized even when incubated 36 days indicating these cells are unable to proliferate in the absence of attachment to a solid surface. Additionally, KSHV-infected ECFCBLs showed no indication of multicellular colonies in soft agar. Interestingly, after four weeks in soft agar culture, many multi-cellular colonies were found in the wells containing K-ECFCLYs indicating the ability of KSHV infected lymphatic ECFCs to grow in an anchorage independent fashion. Scanning and quantifying multiple experiments indicated that approximately 10% of the KSHV infected ECFCLYs formed multicellular colonies (Fig 3B).

The capacity of KSHV-infected ECFCLYs to grow in soft agar was also addressed using the ECFCLYs isolated as an adherent subpopulation from additional donors. To this end, mock- and KSHV-infected ECFCLYs from four different donors were embedded as single cells in soft agar seven days post infection. A fully transformed renal cell carcinoma cell line stably infected with KSHV, iSLK.219 [34], was used as a positive control. We then monitored the growth and formation of small colonies of cells over the course of four weeks. Fig 3C shows images of cells from donor 1 after four weeks of growth. The colonies were calculated and quantified as shown in Fig 3D and 3E. We could not detect any evidence of multicellular colonies in the non-infected samples, however, after four weeks in soft agar, several colonies were observed in the wells containing K-ECFCLYs. Differences in colony number and size are seen from different donors. Additionally, ECFCLYs infected seven days (Fig 3C–3E) before embedding in soft agar showed higher number of cells in colonies when compared to cells embedded after 48h of infection (Fig 3A and 3B), most likely allowing more phenotypic changes to occur in the K-ECFCLYs. These results further support the ability of KSHV infected lymphatic ECFCs to proliferate in an anchorage independent fashion.

Together, this suggests that KSHV may promote the survival and proliferation of lymphatic ECFCs but not blood ECFCs in soft agar. However, this effect was limited to small colonies containing approximately 3–17 cells. Fully transformed cells like iSLK.219 form robust colonies with 100s of cells (Fig 3C and 3D). Therefore, KSHV allows significant but limited proliferation of cells in soft agar indicating minimal transformation of the cells. The slow expansion of K-ECFCLYs in soft agar as compared to the iSLK.219 cell line would be consistent with the slower growth of KS tumors compared to many more aggressive tumors like renal cell carcinoma, the source of SLK cells.

## Gene expression changes induced by KSHV infection of lymphatic ECFCs

To determine the effects of KSHV infection on blood and lymphatic ECFCs, we measured the global gene expression differences between these cells upon primary KSHV infection. mRNA from blood and lymphatic ECFCs that were mock- or KSHV-infected, was analyzed by high throughput RNA sequencing 48 hours post infection. We also compared these gene expression profiles to previous data we generated from mature BECs and LECs [14]. S3A Fig shows a Venn diagram comparing all genes upregulated during KSHV infection by at least 2-fold in all cell types. Approximately 125 genes are specific to KSHV-infected BECs (both mature and ECFCs), while 52 genes were specific to LECs. To validate the RNA-sequencing results, we analyzed selected genes that had differential expression between blood and lymphatic ECFCs

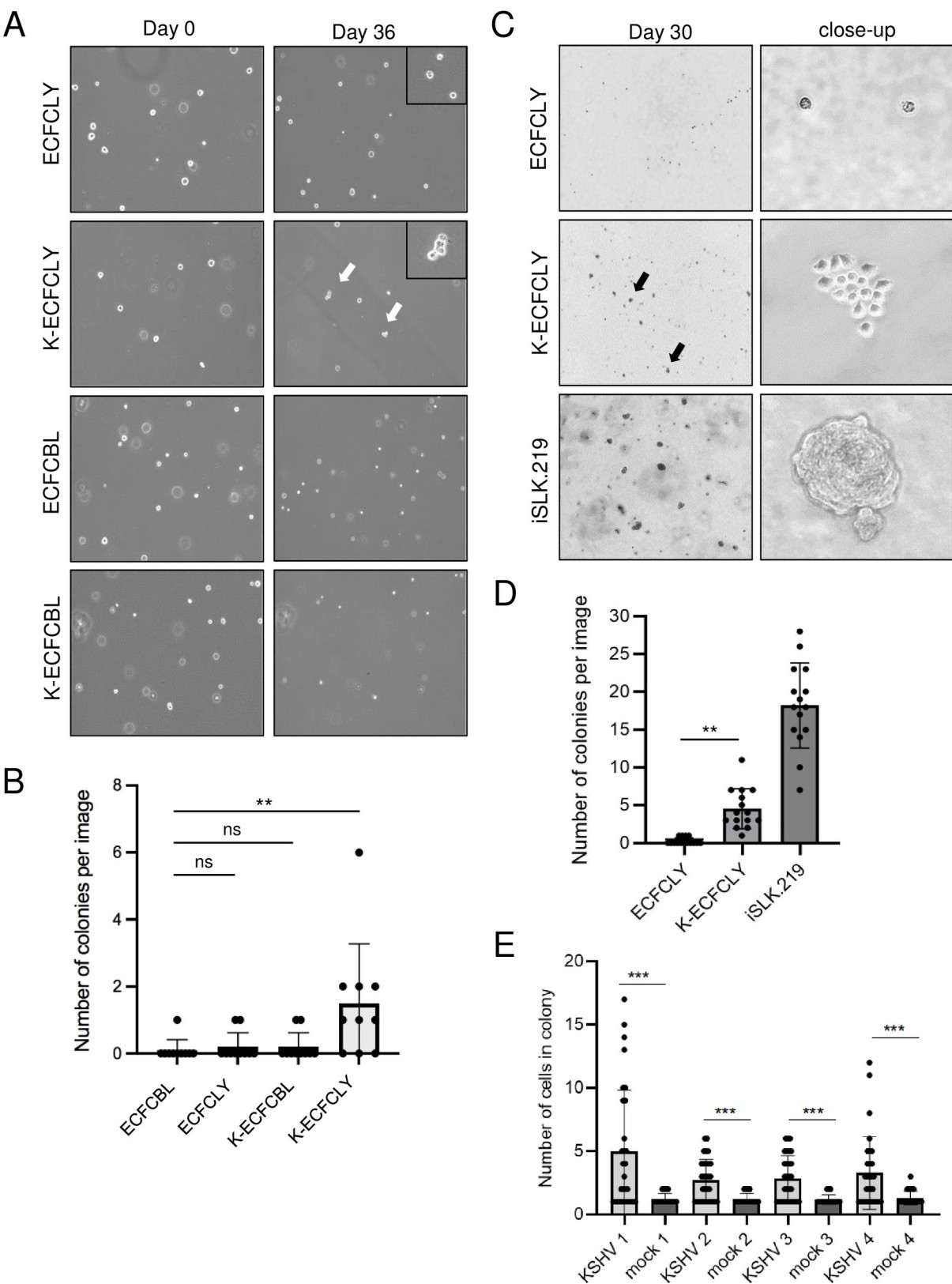

**Fig 3. KSHV confers a survival advantage to lymphatic ECFCs grown in soft agar. A.** Single cells suspensions of mock or wtKSHV-infected lymphatic and blood ECFCs were embedded 48 h.p.i in soft agar overlaid with 0.5ml growth media, and media was replenished every 2–4 days. Images were taken 36 days post plating. **B.** The number of colonies with two or more cells at 36 days post plating were quantified. These experiments were performed up to eight times with similar results. **C.** Lymphatic ECFCs mock- or rKSHV.219-infected were embedded 5 d.p. i. in soft agar. Stably KSHV-infected iSLK.219 was used as a positive control cell line. Images were taken at 30 days post plating. **D.** Quantification of the number of colonies consisting of at least 3 cells from the samples in C. **E.** Quantification of the number of colonies at 30 days post plating from all four different ECFCLY donors infected with rKSHV.219 and embedded 5 d.p.i. in soft agar. Experiments were performed at least three times with similar results. ** p < 0.01, *** p < 0.001.

using custom PrimePCR plates (S3B Fig). These data are consistent with the gene expression changes identified by high-throughput sequencing. The RNA-sequencing reads for both K-ECFCBL and K-ECFCLY were also aligned to the KSHV genome to determine whether KSHV gene expression varies between the two cell types. S3D Fig shows a volcano plot of the KSHV gene expression comparing K-ECFCBL and K-ECFCLY. Only one gene, K15, was shown to be differentially expressed (approximately 2-fold higher expression in K-ECFCLY) suggesting that under these conditions, where the cells are regularly split and constantly proliferating, these cells had similar relative KSHV gene expression profiles.

Using Cytoscape software and the Gene Ontology classification application BINGO, we determined the Gene Ontology terms that were highly enriched among the K-ECFCBLs and K-ECFCLYs. The most highly enriched categories for blood ECFCs are listed in Table A in S1 Text. Interestingly, these include many immune response categories. The individual genes for some of these categories are listed in Table B in S1 Text. In contrast, lymphatic ECFCs were not enriched in immune response categories, but were enriched in genes involved in cell differentiation and signaling (Tables A and B in S2 Text). Additionally, we used the Broad Institute Gene Set Enrichment Analysis to compare gene sets commonly enriched in KSHV-infected lymphatic cell types (K-LEC and K-ECFCLY) to determine whether they share similar profiles. As seen in S3 Text, several of the top-rated gene sets are common between the two lymphatic cell types, such as TNF-α signaling and p53 pathway members. To confirm whether genes involved in the innate immune response are upregulated in KSHV-infected blood ECFCs but not lymphatic ECFCs, we performed real-time quantitative RT-PCR on RNA from mock and KSHV infected blood and lymphatic ECFCs. S3C Fig shows that several genes involved in innate immunity, such as viperin and IFI6, are induced by KSHV in blood ECFCs but not in lymphatic ECFCs. This suggests that lymphatic and blood ECFCs may have differential innate immune responses to KSHV infection similarly to neonatal LECs and BECs [35]. The differentially expressed genes between KSHV-infected blood and lymphatic ECFCs are listed in S1 Table.

High-throughput RNA sequencing was also carried out with lymphatic ECFCs, isolated as adherent subpopulation from a different donor (Donor 1; S2 Fig) as in the above-described analysis. Cells were mock- or KSHV-infected for seven days before total RNA was harvested for sequencing. The gene expression profiles were compared to the data obtained from lymphatic ECFCs infected with KSHV for 48h as described above. S3E Fig shows a Venn diagram of 1444 shared gene expression changes between the two infected ECFCLY-isolates, also listed in S2 Table. The higher number of differentially expressed genes with ECFCLYs infected for seven days is most likely due to the longer exposure to virus thus allowing more phenotypic changes to occur.

S3F Fig shows a Volcano plot of differentially expressed genes between mock and KSHV-infected ECFCLY after seven days of infection. Interestingly, significantly upregulated genes include those involved in lymphatic endothelial specification, such as LYVE1, Podoplanin, and VEGFC. Also, some mesenchymal genes implicated in EndMT were moderately, but significantly, upregulated. Selected genes were validated by RT-qPCR shown in S3G Fig, and they are consistent with the gene expression changes identified by RNA sequencing. Although the

increase in SOX18 expression upon KSHV infection was prominent at the protein level (Fig 1D), it was only modest at the mRNA level (less than 2-fold), and not reaching the threshold of differential expression chosen for the genes included in the Volcano blot (S3F Fig).

## SOX18 inhibition reduces the hallmarks of KSHV infection in lymphatic ECFCs *in vitro*

Our recent data demonstrated that SOX18, expressed abundantly in KS tumors and K-LECs, binds to the origins of KSHV replication and increases the intracellular viral DNA genome copies [12]. To address the SOX18 role in K-ECFCLYs the cells were treated with siRNA targeting SOX18 or non-targeting control siRNA (siNeg) after five days of infection and analyzed three days later for KSHV genome copies, number of LANA positive cells and infectious virus release on naïve U2OS cells (Fig 4A–4D). Similar to K-LECs, depletion of SOX18 significantly reduced the intracellular viral genome copies, number of LANA-positive cells and the virus titers (Fig 4A–4D). To measure the effect of SOX18 inhibition by the SM4 inhibitor on lymphatic K-ECFCLYs, GFP and RFP positive cells (to monitor latent and lytic infection), spindling phenotype of the cells, in addition to the above-mentioned phenotypes were measured as the hallmarks of KSHV infection. K-ECFCLYs (at 5 days p.i.) were treated with SM4 for six days at concentrations of 1 μM, 25 μM and 50 μM and using corresponding concentrations of the vehicle, DMSO as controls. Fig 4E shows that similar to depletion of SOX18 by siRNA, 25 μM of SM4 was sufficient to reduce the GFP positive cells, spindling phenotype and RFP signal indicating cells undergoing lytic cycle. Potential toxicity of SM4 treatment was evaluated after a 6-day treatment by Trypan Blue staining. As shown in Fig 4E (bottom panel) and Fig 4F (top panel), 50 μM of SM4 induced some toxicity in the infected cells, however, this was not observed in the uninfected ECFCLYs (Fig 4F, middle and bottom panels). This finding indicates higher sensitivity of KSHV-infected cells for SM4, possibly due to the high SOX18 protein expression (Fig 1D). Furthermore, SM4 significantly, and in a dose-dependent manner, reduced the number of KSHV genome copies, LANA positive cells and released infectious virus from the infected lymphatic ECFCs (Fig 4G–4I). These data demonstrate that SOX18 is required to maintain the high number of intracellular KSHV genomes and viral titers in the K-ECFCLYs, similar to K-LECs [12].

By using a three-dimensional (3D), organotypic cell model for KSHV infection we have previously shown that KSHV promotes the ability of mature lymphatic endothelial cells to invade into 3D cross-linked fibrin [10]. Next, we tested if KSHV infection would induce the 3D sprouting growth of the lymphatic ECFCs and if it was dependent on SOX18 function. To this end, mock- and KSHV-infected lymphatic, and blood ECFCs were first allowed to form spheroids overnight and then embedded in 3D fibrin. As shown in Fig 5A, KSHV induced outgrowth of sprouting cells from the lymphatic ECFC spheroids, which was completely abolished during the 6-day 25 μM SM4 treatment. On the contrary, both mock- and KSHV-infected blood ECFC spheroids were sprouting (Fig 5B and 5D), indicating the intrinsic, angiogenic properties of the blood ECFCs, and not induced by the virus. Fittingly, SM4 had no effect on the KSHV-infected blood ECFCs, which also do not express SOX18 (Figs 1D and S1B). While 25 μM SM4 was sufficient to completely inhibit sprouting of K-ECFCLYs, we also used the higher concentration of 50 μM on the SOX18-expressing, mock and KSHV-infected ECFCLYs to assess potential toxicity in 3D, which had been seen in 2D (Fig 4F). Mock-infected lymphatic ECFCs spheroids did not sprout and neither 25 μM nor 50 μM SM4 had any effect on their morphology (Fig 5C). Together these data support that the lymphatic ECFCs represent a viable model for KSHV infection and testing of the translational potential of novel targets for KS.

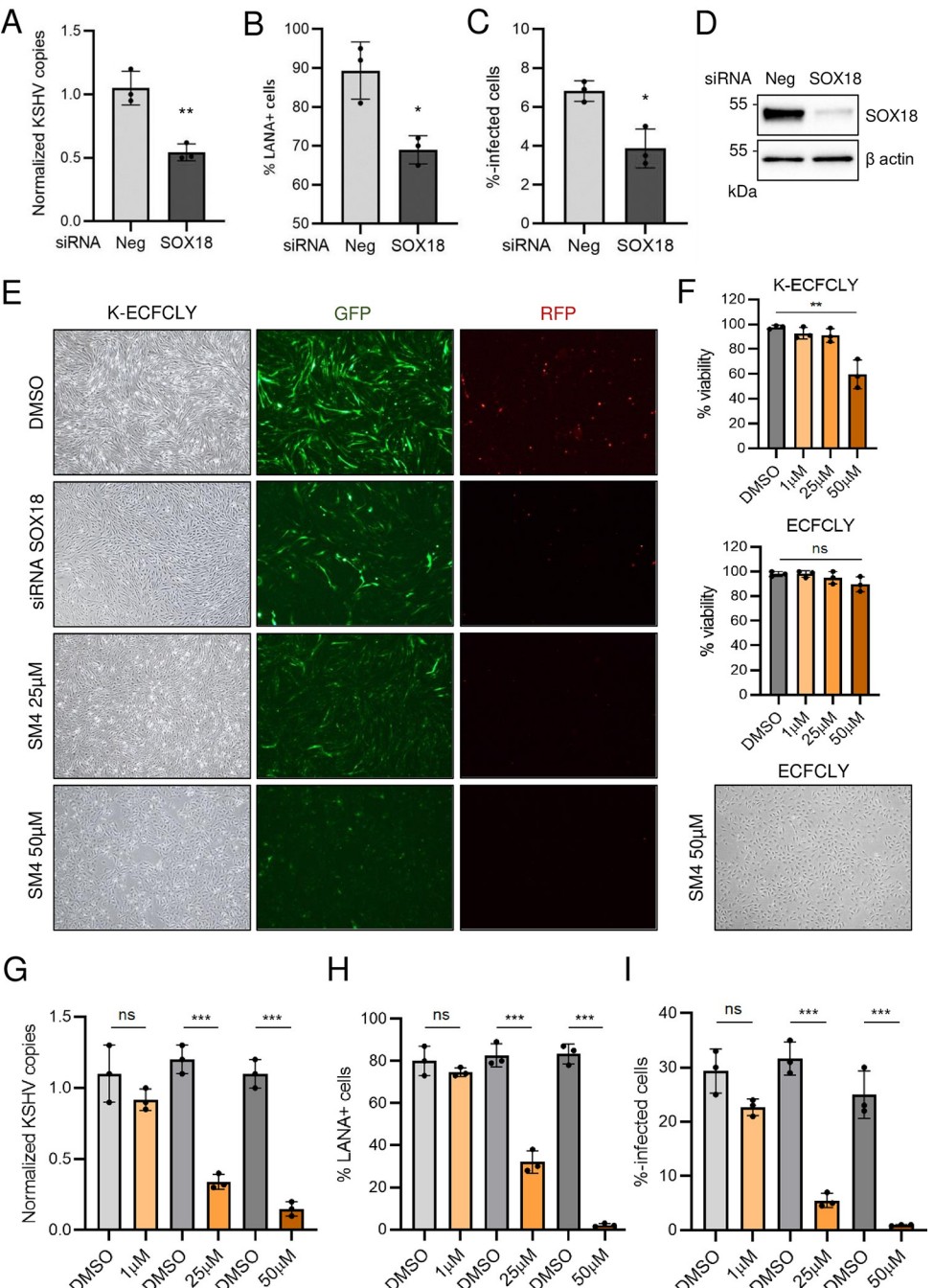

**Fig 4. SOX18 inhibition reduces the hallmarks of KSHV infection in lymphatic ECFCs *in vitro*.** Lymphatic ECFCs were infected with KSHV for 5 days, and treated with siRNA targeting SOX18, or control siRNA (siNeg) transfected for 72h (**A-E**) or with the indicated concentrations of SM4 inhibitor or the corresponding concentrations of DMSO control for 6 days, replenished at day 3 (**E-I**). The effect of treatments is shown as quantified; **A** and **G** of KSHV genome copies from total DNA, **B** and **H** of percent of LANA positive cells on 96-well plate, and **C** and **I** of KSHV titers measured from virus release assay on naïve U2OS cells. **D.** The efficiency of SOX18 silencing shown at a protein level. **E.** Changes in spindling morphology, GFP (latent) and RFP (lytic) cells shown with the siRNA targeting SOX18 or 25 μM and 50 μM of SM4 and the highest DMSO concentration control. * p < 0.05, ** p < 0.01, *** p < 0.001.

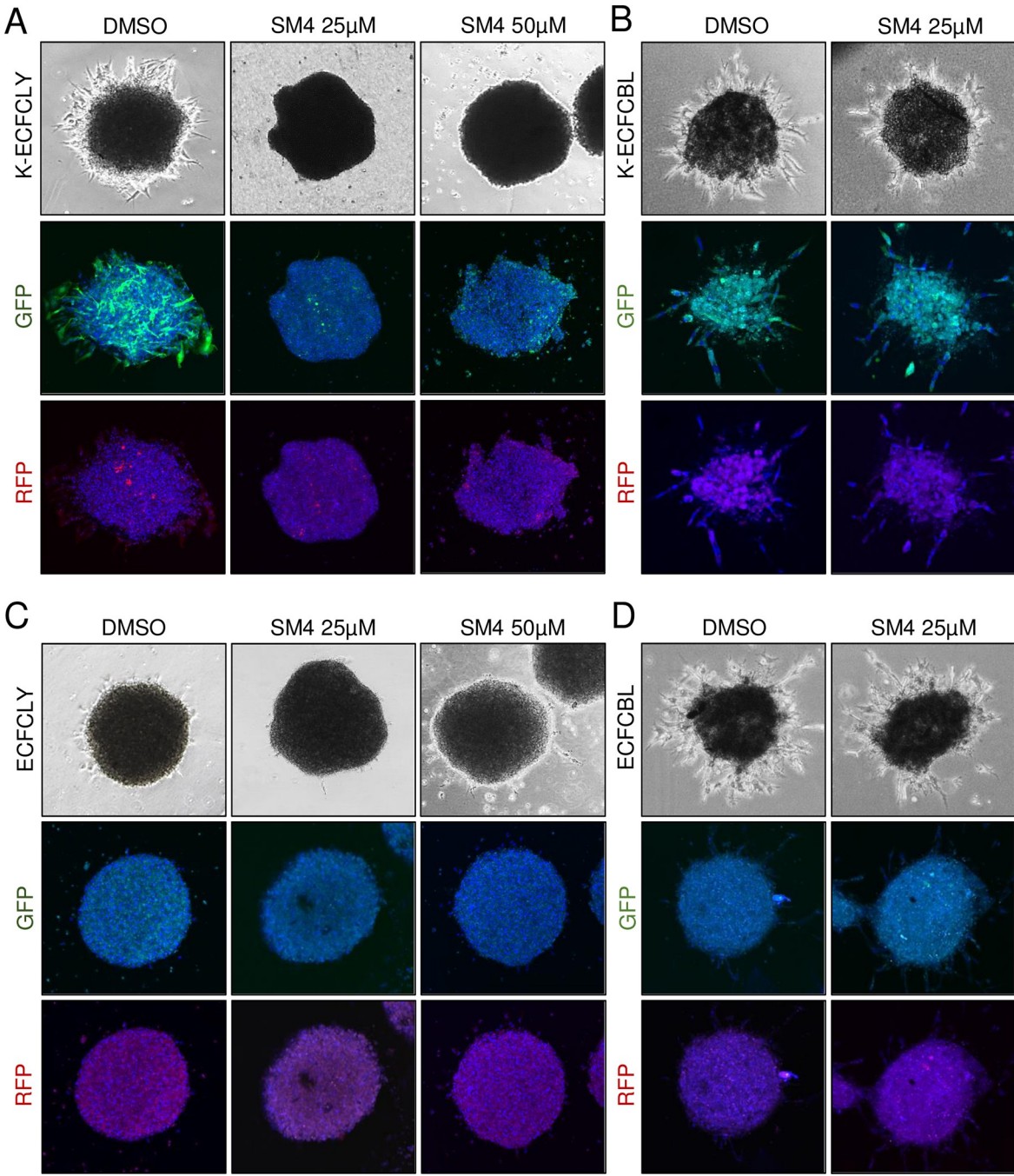

**Fig 5. SOX18 inhibition reduces the KSHV-induced sprouting in lymphatic ECFCs.** Lymphatic (**A** and **C**) and blood (**B** and **D**) ECFCs were either mock or KSHV-infected, allowed to form spheroids overnight and then embedded in 3D fibrin. Spheroids were treated for six days with either 25uM or 50 μM of SM4, and the highest DMSO concentration control, fixed and imaged. Phase contrast of live spheroids and confocal images of fixed and stained spheroids are shown at treatment day 6. These images do not necessarily always represent the same spheroid.

## SOX18 inhibition reduces the hallmarks of KSHV infection *in vivo*

As the KSHV-infected lymphatic ECFCs support the complete lytic replication program, proliferate, and show emerging transformation, we decided to test their long-term persistence *in vivo* in NSG mice when implanted subcutaneously (Fig 6A). Mock and rKSHV.219-infected

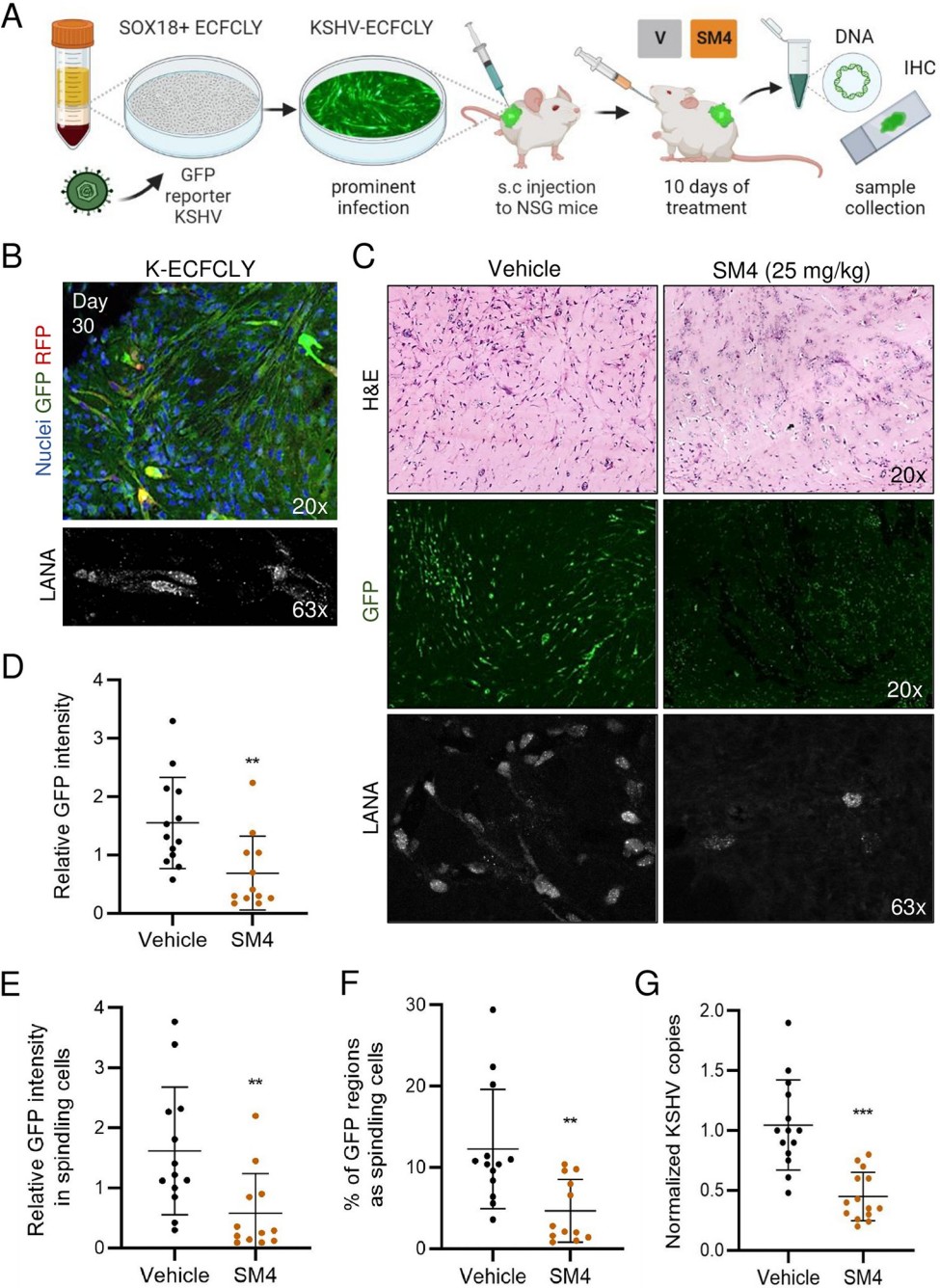

**Fig 6. SM4 inhibits the hallmarks of KSHV infection in lymphatic ECFCs in vivo. A.** Schematic of the experimental in vivo layout of SOX18 inhibition on ECFCLY KSHV-infection model and sample collection. **B.** Lymphatic ECFCs infected with rKSHV.219 for seven days, or left uninfected, were implanted subcutaneously into NSG mice. Long-term persistence of K-ECFCLYs is seen as spindling GFP-positive cells and nuclear LANA signal, from grafts extracted 30 days post implantation. **C-E.** Lymphatic ECFCs and LECs, infected with KSHV for seven days, or left uninfected, were implanted subcutaneously to NSG mice in a ratio of 90–95% of K-ECFCs and 5–10% of K-LECs. 24h post implantation the mice were treated with a daily dose of either SM4 or Vehicle control for 10 days after which the mice were sacrificed, and grafts collected for total DNA and IHC. **C.** H&E staining of the sections from the grafts imaged by a slide scanner. GFP and LANA imaged by a confocal microscope. Quantification of **D** overall GFP intensity, **E** GFP intensity only in spindling cells and **F** spindling morphology (n = 12/ SM4 and 13/ Vehicle group). **G.** Normalized KSHV genome copies from total DNA measured with RT-qPCR between SM4 or Vehicle treated mice (n = 14/group). ** p < 0.01, *** p < 0.001.

lymphatic ECFCs were implanted subcutaneously in Matrigel after seven days of infection (7 d.p.i.). After a 30-day period, visible cell plugs were collected and analyzed for the presence of KSHV DNA. The intensity of GFP (from the rKSHV.219 latent infection) was also quantified and LANA expression verified by IFA and imaging from the paraffin embedded sections. Although no tumorigenic growth was observed, Fig 6B shows that the implanted K-ECFCLYs survived for 30 days in the NSG mice, recapitulated the spindling phenotype of KS tumor cells in the lesions and expressed GFP and LANA. Mock-infected ECFCLYs did not show a spindling phenotype or express GFP (S5A Fig). This suggests that K-ECFCLYs represent a promising *in vivo* model to investigate the potential treatment modalities for KS.

We next tested the effect of SM4 in this ECFCLY model (Fig 6A) except that a small amount (5–10%) of K-LECs were mixed with K-ECFCLYs to provide spontaneously lytic cells that could contribute to the inflammatory microenvironment and produce more infectious virus. Using this K-LEC/K-ECFCLY mixture we first confirmed that the effect of 25 µM SM4 on KSHV genome copies, LANA positive cells and released infectious virus was similar to what we have seen in 100% K-ECFCLY *in vitro* (S4 Fig). Next, $4 \times 10^6$ mock- or rKSHV.219 infected ECFCLYs, and LECs, were injected as a LEC/ECFCLY mixture subcutaneously into NGS mice 7 d.p.i. The final proportions of the cells were 90–95% of K-ECFCLYs and 5–10% of K-LECs. One day after implantation, SM4 or vehicle (80% PEG-400, 10% Solutol HS-15, 10% dH2O) was administered orally at a dose of 25 mg/kg of body weight daily for 10 sequential days as described in [27], after which the cell plugs were collected and analyzed by IHC for cell morphology, IFA for KSHV infection, and qPCR for KSHV DNA.

Interestingly, IHC of the infected cell plugs from mice treated with SM4 displayed several necrotic areas in the grafts (Fig 6C, top right panel), whereas the vehicle group showed cells with a spindle-like, elongated morphology, often found in clusters (Fig 6C, top left panel). IHC of the mock infected cells treated with SM4 did not show any signs of necrosis (S5B Fig, top right panel). IFA was performed for sections to distinguish the GFP-expressing, KSHV-infected cells with spindle-like morphology. GFP and LANA stained sections imaged with a confocal microscope are shown in Fig 6C (middle and bottom panels). The intensity of GFP signal and the areas of spindling cells in the sections differed significantly between the treatment groups (Fig 6D–6F). While an intense GFP signal in spindling cells was observed in the Vehicle group, the overall GFP signal in the SM4 group was weak, having also significantly less cells with spindling morphology. Quantification confirmed that the Vehicle group had higher GFP intensity and more spindling cells, when normalized to the section area, and when compared to the SM4 treatment group (Fig 6D–6F). The mock infected ECFCLYs did not express GFP, and the images were used to subtract the autofluorescence for quantification (S5B Fig, bottom panels). These results indicate the strong potential of SOX18 inhibition to reduce the hallmarks of KSHV infection in this novel in vivo KSHV infection model.

We next measured the KSHV genome copy numbers by qPCR from the extracted total DNA of the grafted samples. The human repetitive ALU-sequences as an internal control were used as a highly sensitive and specific quantification for the human cells among the abundant rodent cells. The KSHV genome copies were quantified using primers specific for LANA. Fig 6G shows that SM4 significantly decreased the relative KSHV genome copy numbers compared to the Vehicle control group further supporting the potential of SOX18 inhibition as a viable therapeutic modality for KS.

## Discussion

Studies of KS tumors are limited by the lack of a clear understanding of the source of the main proliferating tumor cell, the spindle cell. While spindle cells express markers of lymphatic

endothelium and most closely align to the gene expression pattern of lymphatic endothelial cells, they also express blood vascular endothelial and mesenchymal cell markers. However, in contrast to angiogenic endothelial cells, quiescent endothelial cells in mature vessels generally do not migrate. There is evidence that the cells that seed KS tumors can traffic from the kidney as KS of donor origin formed to the lower extremities of the recipient following a kidney transplant from a KSHV-positive donor [18]. Interestingly, endothelial precursor cells traffic throughout the body through the blood and are known to traffic to sites of angiogenesis. Thus, it has been proposed by multiple groups that endothelial precursor cells are a likely source of KS spindle cells [19–21]. Therefore, we sought to isolate the precursor ECFCs and characterize KSHV infection in those cells. When isolating ECFCs from human blood, we found that the standard endothelial colony forming cells could be separated into isolates that expressed blood endothelial markers and isolates that expressed lymphatic specific markers. We also isolated ECFCs from four donors as an adherent subpopulation without clonal selection and expansion of individual colonies. Interestingly, these proliferative isolates represented uniform cultures of predominantly lymphatic ECFCs after a couple of passages. This might indicate that lymphatic ECFCs have a growth advantage over the blood ECFCs and take over the culture when cells proliferate during colony formation.

Like the neonatal blood and lymphatic endothelial cells isolated from foreskins, the lymphatic ECFCs were more permissive to KSHV infection and the viral episomes were maintained over multiple passages at higher levels as compared to the blood ECFCs. KSHV infection did not alter the ability of either cell type to form capillary-like structures in Matrigel indicating that KSHV does not dramatically alter the angiogenic potential of either cell type. However, infection with KSHV slightly decreased the proliferation rate of both blood and lymphatic ECFCs. We have also noted a decrease in proliferation of neonatal and adult endothelial cells [12], so this appears to be a common phenotype of KSHV infection of endothelial cells in general. However, lymphatic ECFCs proliferated at a significantly higher rate than LECs, and KSHV-infection conferred the ECFCLYs a survival advantage, not observed with K-ECFCBLs or K-LECs, as indicated by increased viability of K-ECFCLYs under limiting growth conditions.

To determine if there was any evidence of transformation of the ECFCs despite the reduced proliferation, we examined the proliferation of non-infected and infected blood and lymphatic ECFCs following detachment from a solid matrix. Interestingly, neither non-infected or infected blood ECFCs proliferated in soft agar nor did the uninfected lymphatic ECFCs. However, in a large number of studies using six different isolates of KSHV-infected ECFCLYs, approximately 10% of the cells formed small colonies following incubation of over four weeks in soft agar. While the colonies did not expand beyond the small 3–17 cell colonies, in all the experiments performed we did not detect any colony formation in the non-infected blood or lymphatic ECFCs or the infected blood ECFCs indicating that this proliferation only occurred in the KSHV-infected lymphatic ECFCs and was reproducible. While there were numerous colonies formed, not every cell formed a colony. Thus, the transformation phenotype was limited while a similar assay with iSLK.219, a renal cell carcinoma cell line stably infected with a recombinant KSHV, led to very large colonies in soft agar of hundreds of cells or more. The iSLK.219 cells also formed colonies more rapidly. Of note, in line with our findings, KS tumors are fairly indolent and do not grow rapidly.

To begin to decipher why there are differences in the behavior of lymphatic and blood ECFCs following KSHV infection we performed RNAseq analysis of non-infected and KSHV-infected lymphatic and blood ECFCs. We identified 758 genes that are enriched in KSHV-infected blood ECFCs compared to lymphatic ECFCs (S1 Table). In contrast, 1,395 genes are enriched in KSHV-infected lymphatic ECFCs. Interestingly, when we analyzed the lists of genes for enrichment of Gene Ontology categories, we found that many of the genes induced by KSHV in blood ECFCs were involved in the innate immune response. These include MX1

and MX2, which are interferon-induced genes that are viral restriction factors. MX2 in particular has been shown to be effective at inhibiting herpesviruses, including KSHV [36,37]. Other interferon response genes are also enriched in KSHV-infected blood ECFCs such as IFI6 and RSAD2. It is unclear what effect this enhanced interferon response has on blood ECFCs. However, it could play a role in the reduced susceptibility of blood ECFCs to KSHV infection or the increased rate of episome loss. We previously found that in neonatal LECs the stimulator of interferon genes, STING was not activated [35]. Further work is necessary to determine if the same is true for lymphatic ECFCs. In contrast to the blood ECFCs, genes upregulated by KSHV in lymphatic ECFCs were found to be involved in lymph vessel development and intracellular signaling, in addition to some EndMT factors. What role these genes have on KSHV infection of lymphatic ECFCs remains to be elucidated. We also found that, under high proliferation conditions, the expression of KSHV viral genes was similar between blood and lymphatic ECFCs. The only gene that was determined to be statistically different was K15, which was expressed approximately 2-fold higher in the ECFCLYs. K15 expression is normally restricted to the lytic replication cycle, however, its similarity to the EBV protein LMP2A suggests that the cell-type specific expression of K15 during latency in ECFCLYs could play a role in differences between ECFCBLs and ECFCLYs. Further studies on K15 gene expression in ECFCLYs are necessary to determine its functional significance.

We recently reported high SOX18 expression in KS tumors and that SOX18 was required to maintain the high number of intracellular KSHV genome copies in infected LECs, suggesting SOX18 as an attractive therapeutic target for KS [12]. Similar to K-LECs we found that KSHV-infected lymphatic ECFCs express high levels of SOX18 and inhibition of SOX18 by a specific small molecule inhibitor SM4 significantly reduced the intracellular viral genome copies also in KSHV-infected ECFCLYs. SOX18 inhibition with SM4 at 25 μM was sufficient to reduce the KSHV hallmarks without toxicity to the cells. Interestingly, higher concentration of SM4 (50 μM) induced toxicity to the infected cells, but not to the uninfected cells. This finding indicates the specific sensitivity of SM4 in KSHV infected cells, which express high levels of SOX18. Accordingly, SOX18 inhibition with 25 μM of SM4 was sufficient to abolish the KSHV-induced capacity of K-ECFCLYs to sprout into the 3D cross-linked fibrin, whereas it had no effect on the KSHV-independent, angiogenic sprouting of the SOX18-negative ECFCBLs.

Since K-LECs do not support long-term growth in culture and do not show emerging signs of transformation they cannot be used in preclinical in vivo studies to test the efficacy of SOX18 inhibitors. Therefore, we investigated here whether KSHV-infected lymphatic ECFCs engrafted into NSG mice could serve as a physiologically relevant infection model to test the potential of SOX18 inhibition as a therapeutic modality for KS. The KSHV-infected ECFCLYs survived as xenografts in mice for at least a month and recapitulated the hallmarks of KSHV infection, which were significantly reduced upon SOX18 inhibition by SM4. Intriguingly, a recent publication reported a 6-month oral propranolol treatment of a patient with classic KS resulting in a substantial decrease in size of KS lesions associated with a reduction in both vascular proliferation and KSHV infection [38]. Our findings of SOX18 inhibition using the novel K-ECFCLYs *in vivo* model and the success of propranolol, an FDA-approved drug and a SOX18 inhibitor, in treating classic KS encourage further investigations on the use of specific SOX18 inhibitors as a future KS therapy [39].

Together our data suggest that lymphatic ECFCs are more susceptible to KSHV infection and may have increased survival and proliferation capabilities during infection. While this does not definitively solve the issue of the source of spindle cells, it shows that infected lymphatic ECFCs are certainly a possible source. Moreover, our data supports that lymphatic ECFCs represent a viable new in vivo model for KSHV infection, suitable for translational

studies testing new therapeutic approaches for KS. Further work to analyze KSHV infection of lymphatic ECFCs is warranted.

## Materials and methods

### Ethics statement

The blood samples for the isolation of ECFCs were obtained from healthy volunteer donors through the Puget Sound Blood Bank (Seattle, WA, USA) and the Finnish Red Cross Blood Services (Helsinki, Finland) under granted permissions. No data that can lead to the identification of the donors will be available to us. The maintenance and all procedures with the mice were performed in authorized facilities, at the Laboratory Animal Center, HiLIFE, University of Helsinki (Finland), by trained certified researchers, and under a license approved by the national Animal Experiment Board, Finland (license number ESAVI/10548/2019 for tumor growth and ESAVI/22896/2020 for the oral gavage administration).

### Cells

For isolation of clonal populations of ECFCs the blood samples were obtained from the Puget Sound Blood Bank (Seattle, WA, USA) as either outdated whole blood (approximately 500 ml per unit) or leukocyte reduction filters, which yielded between 4-7x10$^8$ leukocytes after elution. Whole blood was mixed 1:2 with phosphate-buffered saline (PBS) containing 10,000 units per liter heparin and 0.02% EDTA. Cells from leukocyte reduction filters were eluted with 200 ml PBS containing 10,000 units per liter heparin and 0.02% EDTA. ECFCs were then isolated and cultured as previously described [40]. Briefly, mononuclear cells (MNCs) were isolated by density gradient centrifugation using Histopaque-1077 (#10771; Sigma-Aldrich, St.Louis, USA). Cells from the buffy coat layer were seeded onto tissue culture dishes coated with 50 μg/mL rat tail collagen I (#CB-40236; Corning, NY, USA) at approximately 5x10$^7$ cells/well of a 6-well dish. Cells were cultured in EGM-2 media (#CC-3162; Lonza, Walkersville, USA) supplemented with an extra 5% FBS (for a total of 10%) and 1% penicillin/ streptomycin/ amphotericin B (#15240062; Gibco). Media was replaced daily for seven days and subsequently every two days until colonies appeared. Individual colonies of cells were isolated with cloning rings and clonally expanded. All experiments were performed on cells (USA) between passage 8 and 15.

For isolation of an adherent subpopulation of ECFCs, fresh blood samples were obtained from the Finnish Red Cross Blood Services (Helsinki, Finland) as buffy coat concentrates devoid of platelets. After dilution of 1:2 in PBS (#21-040-CV; Corning), the cells were isolated according to manufacturer's instructions using SepMate tubes (#146138; STEMCELL Technologies, Cambridge, UK) and Cytiva Ficoll-Paque PLUS density gradient media (#45-001-749; Fisher Scientific). The final number of mononuclear cells separated from each donor's blood bag was approximately 1.5x10$^7$ cells/mL, yielding a total of 200 million cells. The cells were transferred onto a fibronectin 25 μg/mL (#F0895; Sigma) and 15 μg/mL rat tail collagen (#7661; Merck, St.Louis, USA) pre-coated cell culture 6-well plate with EBM-2 basal endothelial media (#CC-3156; Lonza) supplemented with 10% FBS and a EGM-2 MV growth factor bullet kit (#CC-4147; Lonza) 3 mL/well. Cultures were followed up to three weeks, washing away any non-adherent cells and changing fresh supplemented EBM-2 media every other day. When populations of adherent, cobblestone-like cells resembling endothelial cells formed, they were split with Trypsin-EDTA (#ECM0920D; Euroclone) onto pre-coated 10 cm culture dishes without colony selection, meaning that all adherent cells were propagated. These cells were then frozen in Cryo-SFM freezing media (#C-29912; Promocell, Heidelberg, Germany) and stored in a liquid nitrogen tank prior to the phenotype analyses and using as an infection model. All experiments were performed on cells (FIN) between passage 3 and 8.

Primary human dermal lymphatic and blood endothelial cells (LEC #C-12216 and BEC #C-12211; Promocell) and neonatal dermal microvascular endothelial cells (hDMVEC) (Lonza) were maintained as monolayer cultures in EBM-2 medium (Lonza) supplemented with 5% fetal bovine serum, vascular endothelial growth factor, basic fibroblast growth factor, insulin-like growth factor 3, epidermal growth factor, and hydrocortisone (EGM-2 media). Human osteosarcoma cell line U2OS (ATCC #HTB-96) is highly susceptible to KSHV infection and was therefore used as a naïve cell line to determine KSHV titers and the efficacy of infection in the virus release assays. iSLK.219 [34] is an RTA -inducible renal-cell carcinoma SLK cell line, stably infected with a recombinant KSHV.219. U2OS and iSLK.219 were grown in DMEM (#ECB7501L; BioNordika), supplemented with 10% FBS (#10270–106; Gibco), 1% L-glutamate (#ECB3000D; BioNordika), and 1% penicillin/streptomycin (#ECB3001D; BioNordika). iSLK.219 cells were also supplied with 10 μg/mL puromycin (#P8833; Sigma), 600 μg/mL hygromycin B (#687010; Invitrogen, Lithuania), and 400 μg/mL Geneticin G418 (#04727878001; Roche, Switzerland). All cells were propagated in a humified incubator at standard conditions and primary cells were used until passage five. Cells were regularly tested negative for *Mycoplasma* (MycoAlert Mycoplasma Detection Kit, #LT07-705; Lonza).

## FACS analysis

The ECFCs isolated as an adherent subpopulation (FIN) were analyzed by FACS using the following fluorescently conjugated antibodies for endothelial surface markers: human anti-Alexa Fluor 647-Podoplanin (#395003) and 647-CD34 (#343617), PE-VEGFR3/FLT-4 (#356203), and PE-CD31 (#303105) or corresponding IgG isotypes Alexa Fluor 647 mouse IgG2a (#400239) and PE mouse IgG1 (#400113) (Biolegend, San Diego, USA). For the nuclear EC markers, cells were fixed and permeabilized with MetOH and stained with rabbit polyclonal anti-PROX1 (rabbit) (#11067-2-AP; Proteintech, Rosemont, USA) and mouse monoclonal anti-SOX18 (#sc-166025; Santa Cruz Biotechnology, Dallas, USA) antibodies, followed by staining with secondary antibodies conjugated to Alexa Fluor 488 and 594 stains (#A11034, #A21203; Invitrogen). All samples were analyzed with BD Accuri C6 flow cytometer using unstained cells and conjugated IgG isotypes, or secondary antibody only treated controls.

## Viruses and infection

Wild-type (wt)KSHV inoculum was obtained from BCBL-1 cells ($5x10^5$ cells/mL) induced with 20 ng/mL of TPA (12-*O*-tetradecanoylphorbol-13-acetate; Sigma). After 5 days, cells were pelleted, and the supernatant was run through a 0.45 μm pore-size filter (Whatman, China). Virions were pelleted at 30,000xg for 2 h at 4˚C in a JA-14 rotor, Avanti-J-25 centrifuge (Beckman Coulter, Palo Alto, USA). The viral pellet was resuspended in EBM-2 without supplements. wtKSHV infections at low, less than 5% of cells infected, and high MOI, enough virus to infect the majority of the cells in the culture, were performed in serum-free EBM-2 supplemented with 8 μg/mL polybrene (#H9268; Sigma) for 3 h, after which the medium was replaced with complete EGM-2 with supplements. Mock infections were performed identically except that concentrated virus was omitted from the inoculum. Virus titers were determined by infecting tert-immortalized microvascular endothelial (TIME) cells with dilutions of each virus preparation, followed by immunofluorescence staining using anti-LANA (rabbit polyclonal) and ORF59 (#13-211-100; Advanced Biotechnologies, Inc.) antibodies at 48 hours post infection.

The concentrated virus preparation of recombinant KSHV.219 virus was produced from iSLK.219 [34] as described in [12] and centrifuged at 22,000 rpm for 2 h at 4˚C in an SW-12 rotor of the Optima XL-80K ultracentrifuge (Beckman Coulter). The concentrated virus was resuspended in ice-cold PBS, snap-frozen and stored at -80˚C. Cells infected with rKSHV.219

express green fluorescent protein (GFP) from the constitutively active human elongation factor 1-α (EF-1α) promoter and red fluorescent protein (RFP) under the control of RTA-responsive polyadenylated nuclear (PAN) promoter, expressed only during lytic replication. Virus titers were determined by infecting naïve U2OS cells using serial dilutions of the concentrated virus and assessing the amount of LANA+ cells 24h post-infection with an automated cell imaging system ImageXpress Pico (Molecular Devices, San Jose, USA). All endothelial cells were always infected with rKSHV.219 at the same low MOI of 1–2 in EGM-2 media with supplements in the presence of 8 μg/mL polybrene (Sigma) and spinoculation at 450g for 30 min, RT with the 5804R centrifuge (Eppendorf, Germany). Mock infections were performed identically except that concentrated virus was omitted from the inoculum.

## Immunofluorescence

ECFCLYs (FIN) and ECFCBLs (USA) were plated on a viewPlate-96black (#6005182; Perkin Elmer, Waltham, USA), and infected with rKSHV.219. Cells were fixed with 4% PFA 5 d.p.i, permeabilized with Triton X-100 (#T9284; Sigma) and stained with antibodies against rat monoclonal anti-HHV-8 LANA (LN-35; #ab4103; Abcam, Cambridge, UK) or mouse monoclonal anti-K8.1 (#sc-65446; Santa Cruz) and Hoechst 33342 (#14533; Sigma). Anti-rabbit Alexa Fluor 488 or anti-rat Alexa Fluor 647 (#A11034, #21247; Invitrogen) were used as secondary antibodies. Nuclei were visualized with Hoechst 33342 (#14533; Sigma). Images were taken using an automated cell imaging system ImageXpress Pico (Molecular Devices) and % of LANA or K8.1 positive cells were quantified using pipeline created in CellProfiler (Broad Institute, Cambridge, USA).

## Immunoblotting

Cell lysis (ECFCLY; FIN, ECFCBL; USA), SDS-PGE and immunoblot were performed as described in [41]. The following primary antibodies were used: mouse monoclonal anti-β actin (#sc-8432; Santa Cruz); mouse monoclonal anti-Vinculin (#sc-73614; Santa Cruz); Mouse monoclonal anti-SOX18 (D-8) (#sc-166025; Santa Cruz); Rat monoclonal anti-HHV-8 LANA (LN-35; #ab4103; Abcam); Mouse monoclonal anti-K8.1 (#sc-65446; Santa Cruz); rabbit monoclonal anti-GFP (a kind gift from J. Mercer; UCL, London, United Kingdom). Following secondary antibodies were used: anti-mouse, anti-rabbit and anti-rat IgG HRP conjugated (#7076, #7074, #7077; Cell Signaling Technology).

## Virus release assay

One day prior to titration, 8x10$^3$ naïve U2OS cells/well were plated on the viewPlate-96black (Perkin Elmer). Cells were spinoculated in the presence of 8 μg/mL of polybrene using a serial dilution of precleared supernatant from the infected LEC, BEC, ECFCLY (FIN) or ECFCBL (USA). 48h later, cells were fixed with 4% PFA, permeabilized with Triton X-100 and stained with antibodies against GFP (a kind gift from J. Mercer; UCL, London, United Kingdom) to detect the rKSHV.219-infected cells or LANA (#ab4103; Abcam) and Hoechst 33342 (Sigma). Images were taken using the automated cell imaging system ImageXpress Pico and KSHV + cells were quantified using pipeline created in CellProfiler.

## siRNA transfections

Transient transfection of siRNA of a semi-confluent culture of KSHV-infected lymphatic ECFCs (FIN) was done using OptiMEM (#31985047; Gibco), 1.5 μl of Lipofectamine RNAi-MAX (#13778075; Invitrogen) and 50 nM siRNA per well in a 6-well plate according to

manufacturer's instructions. Next day cells were supplied with fresh media. The following siR-NAs were used: ON-TARGETplus SOX18 siRNA (#L-019035-00); ON-TARGETplus Nontargeting pool siRNA (#D-001810-10) from Dharmacon (Lafayette, USA).

## Cell proliferation and viability

Blood and lymphatic ECFCs (USA) were mock- or KSHV-infected. 48 hours post infection, $3x10^4$ cells/well were seeded in 6-well dishes and imaged using an IncuCyte live cell imaging system (Essen Bioscience Inc., Michigan, USA). Cell confluence of three replicate wells was determined every hour for the duration of the experiment and two biological replicate experiments were performed.

To compare the proliferation rates of mock- and KSHV-infected lymphatic ECFCs (FIN) and LECs, $5x10^3$ cells/well were plated five days post infection on viewPlate-96black and the next day the cells were treated with 10 μM 5-ethynyl-2'-deoxyuridine EdU (Thermo Fisher, Eugene, OR) for 4 h and fixed in 4% paraformaldehyde in PBS. The proliferating cells were visualized using Click-iT EdU Alexa Fluor 647 (#C10340; Molecular Probes) staining according to manufacturer's instructions, and Hoechst 33342. Images were taken using automated cell imaging system ImageXpress Pico and the portion of EdU-containing cells was quantified with CellProfiler software.

For measuring the viability of mock- and KSHV-infected blood (USA) and lymphatic ECFCs (FIN), and LECs in serum and/or growth factor deprived media over time, CellTiter-Glo (#G7572; Promega, Wisconsin, USA) luminescent viability assay was performed on 96-well plates for 20 min seeded with $5x10^3$ cells/well of each cell type in eight corresponding wells and the luminescence from live cells were measured with FLUOstar Omega microplate reader (BMG Labtech, Mölndal, Sweden). The viability was calculated as an average of luminescent signal from the eight wells and presented as relative values.

Potential toxicity of SM4 and DMSO control treatments to the ECFCLYs (FIN) *in vitro* was evaluated with Trypan Blue staining of cells after six days. First, growth media with possible floating cells was collected, after which adherent cells were detached and collected using Trypsin-EDTA, after which all cells were centrifuged and resuspended. The cell suspension was mixed 1:1 with 0.4% Trypan Blue solution (#T8154; Sigma) and counted in a TC20 automated cell counter (Bio-Rad, Hercules, CA) to acquire % of viable cells.

## Tube formation of endothelial cells

Matrigel (10 mg/ml; #354234; BD Biosciences, Bedford, MA) was applied at 0.5 mL/35 mm in a tissue culture dish and incubated at 37°C for at least 30 min to harden. Mock- or KSHV-infected ECFCBLs and ECFCLYs (USA) were removed using Trypsin-EDTA and resuspended at $1.5×10^5$ cells /mL in EGM-2. Cells (1 mL) were gently added to Matrigel -coated dish. Cells were incubated at 37°C, monitored for 4–24 h, and photographed in digital format using a Nikon microscope. Capillaries were defined as cellular processes connecting two bodies of cells. Ten fields of cells were counted for each condition and the mean and standard deviations were determined.

## Soft-agar assay

Mock- and wtKSHV-infected blood or lymphatic ECFCs (USA $3x10^4$ cells/well) were mixed 48h post infection with 0.4% agarose as single cell suspension in growth medium and plated on top of a solidified layer of 0.5% agarose in EBM-2 with supplements in 6-well dishes. Fresh media was replenished every 2–3 days and wells were imaged each week using a BZ-X800 (Keyence) microscope. Additionally, mock- and rKSHV.219-infected lymphatic ECFCs (FIN; $3x10^4$ cells/well) were mixed seven days post infection with 0.4% agarose as single cell suspension in growth medium and plated on top of a solidified layer of 0.5% agarose in EBM-2 with supplements in

6-well dishes. iSLK.219 was used as a positive control. Fresh media was replenished every 2–3 days and wells were imaged each week using an Eclipse Ts2 (Nikon) fluorescent microscope.

## RNA-sequencing

Total RNA was isolated from mock- and wtKSHV-infected blood or lymphatic ECFCs (USA) using the NucleoSpin RNA kit (#740955; Macherey-Nagel, Düren, Germany). Due to differences in susceptibility to KSHV infection between these cell types, titering infections were performed to determine the amount of virus needed to achieve similar infection rates using immunofluorescence staining for LANA and ORF59. RNA was further concentrated and purified using the RNA Clean and Concentrator kit (#R1017; Zymo Research, Irvine, CA). Purified RNA samples were processed at the Fred Hutchison Cancer Research Center Genomic Resources core facility (Seattle, WA) and sequenced using an Illumina HiSeq 2000. Image analysis and base calling were performed using RTA v1.17 software (Illumina, San Diego, CA). Reads were aligned to the Ensembl's GRCh37 release 70 reference genome using TopHat v2.08b and Bowtie 1.0.0 [42,43]. KSHV reads were aligned to ViralProj14158 Strain GK18 reference genome. Counts for each gene were generated using htseq-count v0.5.3p9. Differentially expressed genes were determined using the R package EdgeR (Bioconductor). Genes were called significant with a |logFC| > 0.585 and a false discovery rate (FDC) of <0.05. Gene Ontology enrichment was performed using Cytoscape and BINGO [44]. Using cytoscape software and the Gene Ontology classification application BINGO, we determined the Gene Ontology terms that were highly enriched among the blood and lymphatic specific expressed genes. The most highly enriched categories for each cell type are listed in S1 Text and S2 Text. Gene Set Enrichment Analysis was also performed using the web app available through the Broad institute (http://www.gsea-msigdb.org/gsea/index.jsp) [43,44]. Additionally, the data discussed in this publication have been deposited in NCBI's Gene Expression Omnibus [45] and are accessible through GEO Series accession number GSE54416 and GSE207589 (http://www.ncbi.nlm.nih.gov/geo/query/acc.cgi?acc=GSE54416 http://www.ncbi.nlm.nih.gov/geo/query/acc.cgi?acc=GSE207589).

Total RNA was isolated from three independent experiments of the mock- and rKSHV.219 -infected (7 d.p.i) lymphatic ECFCs (FIN) isolated as an adherent subpopulation from donor #1 using the NucleoSpin RNA extraction kit (Macherey-Nagel), after which purity and concentration was determined with NanoDrop spectrophotometer (Thermo Scientific). The RNA was sequenced in an Illumina Novaseq 6000 (150 bases, paired end) by Novogene (Cambridge, UK). Original image data was transformed to sequenced reads by CASAVA base recognition. Raw data were cleaned from low quality reads and reads containing adapter and poly-N-sequences in FASTP. Clean reads were mapped to the human genome (GRCh38.p12) using HISAT2 with parameters—dta—phred33. Read counts were generated by FeatureCounts [46]. Differentially expressed genes were determined using the R package DESeq2. The resulting P values were adjusted using Benjamini and Hochberg's approach for controlling FDR. Genes with adjusted P value <0.05 were assigned as differentially expressed. Raw data are deposited in NCBI's Gene Expression Omnibus [45] and are accessible through GEO Series accession number GSE207657 (https://www.ncbi.nlm.nih.gov/geo/query/acc.cgi?acc=GSE207657).

## Quantitative RT-PCR

Total RNA was isolated from cells using the RNeasy Plus Mini kit (#74134; Qiagen, Maryland, USA) or NucleoSpin RNA extraction kit (Macherey-Nagel) according to manufacturer's protocols. Real time quantitative Polymerase chain reaction (RT-qPCR) and custom PrimePCR plates (Bio-Rad) or LightCycler480 PCR 384 multiwell plates (#4ti-0382; 4titude FrameStar, Wotton, UK) were used to validate the RNA-Seq results. Primer sequences used to amplify the

**Table 1. Primers used in this study.**

| Primer | Forward primer | Reverse primer |
|---|---|---|
| LANA | *ACTGAACACACGGACAACGG* | *CAGGTTCTCCCATCGACGA* |
| K8.1 | *AAAGCGTCCAGGCCACCACAGA* | *GGCAGAAAATGGCACACGGTTAC* |
| ALU human | *GGTGAAACCCCGTCTCTACT* | *GGTTCAAGCGATTCTCCTGC* |
| genomic actin | *AGAAAATCTGGCACCACACC* | *AACGGCAGAAGAGAGAACCA* |
| CD31 | *AACAGTGTTGACATGAAGAGCC* | *TGTAAAACAGCACGTCATCCTT* |
| VEGFC | *GCCAATCACACTTCCTGCCGA* | *AGGTCTTGTTCGCTGCCTGAC* |
| LYVE1 | *CTGCATGACACCTGGATGGA* | *AAGGGCTGGAAACAAGGACA* |
| Podoplanin | *CGAAGATGATGTGGTGACTC* | *CGATGCGAATGCCTGTTAC* |
| CD44 | *CCCATCCCAGACGAAGACAG* | *ACCATGAAAACCAATCCCAGG* |
| CD90 THY1 | *TCGCTCTCCTGCTAACAGTCT* | *CTCGTACTGGATGGGTGAACT* |
| aSMA | *GACCCTGAAGTACCCGATAGAAC* | *GGGCAACACGAAGCTCATTG* |
| ZEB1 | *GATGATGAATGCGAGTCAGATGC* | *ACAGCAGTGTCTTGTTGTTGTAG* |
| SNAI1 | *GCATTTCTTCACTCCGAAGC* | *TGAATTCCATGCTCTTGCAG* |
| ETS1 | *GAGCTTTTCCCCTCCCCGGAT* | *TGCCGGGGGTCTTTTGGGAT* |
| ETS2 | *AGGAGTTTCAGATGTTCCCC* | *GTCCCAGAATTGTTGGTGAG* |
| MMP1 | *AGTCCGGTTTTTCAAAGGGAA* | *CCTTGGGGTATCCGTGTAGC* |
| CXCR4 | *GCCAACGTCAGTGAGGCAGA* | *GCCAACCATGATGTGCTGAAAC* |
| IL6 | *GAACCTTCCAAAGATGGCTGA* | *CAAACTCCAAAAGACCAGTGATG* |
| TGFB3 | *TGAGCACATTGCCAAACAGC* | *ACTCAGTGGCAAAGCTAGGG* |
| SOX18 | *CTTCATGGTGTGGGCAAAG* | *GCGTTCAGCTCCTTCCAC* |
| actin | *TCACCCACACTGTGCCCATCTACGA* | *CAGCGGAACCGCTCATTGCCAATGG* |
| GAPDH | *AAGGTGAAGGTCGGAGTCAAC* | *TGGAAGATGGTGATGGGATTTC* |
| Viperin | *GTGAGCAATGGAAGCCTGATC* | *GCTGTCACAGGAGATAGCGAGAA* |
| IFI6 | *CCTCGCTGATGAGCTGGTCT* | *CTATCGAGATACTTGTGGGTGGC* |
| IL1R1 | *AGAGGAAAACAAACCCACAAGG* | *CTGGCCGGTGACATTACAGA* |
| MyD88 | *GCACATGGGCACATACAGAC* | *GACATGGTTAGGCTCCCTCA* |

indicated targets are listed in Table 1. Relative abundances of viral mRNA were normalized by the delta threshold cycle method to the abundance of GAPDH or actin.

## Quantification of intracellular viral genome copies

Total DNA was isolated from cells (ECFCLY; FIN, ECFCBL; USA) using NucleoSpin Tissue Mini Kit (#740952; Macherey-Nagel) and the KSHV genome copies were quantified by qPCR using 2XSYBR reaction mix (#K0223; Fermentas, Massachusetts, USA) and unlabelled primers specific for LANA, K8.1, human ALU sequences, and genomic actin, listed in Table 1.

## SOX18 inhibitor treatments *in vitro*

For the in vitro studies, small molecule SOX18 inhibitor SM4 (#SML1999; Sigma / or a kind gift from Gertrude Biomedical Pty Ltd., Australia) was solubilized in DMSO (#D8418; Sigma) to obtain a stock solution of 25 mM and stored in 4˚C.

For SOX18 inhibition in 3D spheroid cultures mock- or KSHV-infected lymphatic ECFCs (FIN) and blood ECFCs (USA) were seeded into 0.5% agarose precoated, round-bottom 96-well plates (#650180; Greiner Bio-One, Austria) at $4x10^3$ cells per well. After 16–24 h incubation at 37˚C, the preformed spheroids were transferred into the fibrin gel consisting of plasminogen-free human fibrinogen (final concentration 3 mg/mL; #341578; Millipore) and human thrombin (final concentration 2 U/mL; #605190; Merck) in 50 μL Hank's Balanced

Salt Solution HBBS (#14025; Gibco) supplemented with 400 μg/mL aprotinin (#616370; Calbiochem). The gels were cast onto the bottom of 12-well plates (#665180; Greiner) and incubated for 1 h at 37°C to allow complete gelling followed by addition of EGM-2 culture medium. The next day, SM4 or DMSO as a control was added to the culture media, replenished at day 3 and followed until day 6. Phase contrast images were taken with the Eclipse Ts2 (Nikon) microscope and the spheroids were fixed with 4% PFA for 1 h RT. The spheroids were then stained by anti-rabbit GFP, and Hoechst 33342 as described in [10] and analyzed by confocal microscopy.

### *In vivo* model development and SOX18 inhibition

Female NSG mice (Nonobese diabetic (NOD)/severe combined immunodeficiency (SCID); NOD.Cg-Prkdc^scid Il2rg^tm1Wjl/SzJ) used in this study were provided by Jackson Laboratory and acquired through Scanbur (Germany). The mice were acclimatized for 7 days in isolation. After the isolation period, mice were trained for handling, weighing and finally to oral gavage tube feeding with clean water to reduce stress for the animals during the experimental procedures. The maintenance and all procedures with the mice were performed in authorized facilities, at the Laboratory Animal Center, HiLIFE, University of Helsinki (Finland), by trained certified researchers, and under a license approved by the national Animal Experiment Board, Finland (license number ESAVI/10548/2019 for tumor growth and ESAVI/22896/2020 for the oral gavage administration).

When the majority (about 90%) of the rKSHV.219 -infected lymphatic ECFCs (FIN) expressed GFP, with some (about 5%) expressing RFP, the cells were collected and mixed with K-LECs, almost fully infected with rKSHV.219, with a substantial part (20–30%) expressing RFP. The combined cell preparation consisted of 90–95% of KSHV-infected lymphatic ECFCs and 5–10% of K-LECs. K-LECs were included to provide the more spontaneously lytic cells that can contribute to the inflammatory microenvironment and produce more infectious virus than the ECFCs. $5 \times 10^6$ cells/ 100 μL of cells were embedded in media containing ice-cold growth-factor reduced Matrigel (#356231; Corning, NY, USA) at 3 mg/mL concentration. 100 μL of the cell-Matrigel suspension was injected subcutaneously to both sides of the flanks of NSG mice, using total of 14 mice/ group. Additionally, mock infected ECFCLYs and LECs were mixed in the same ratio and implanted into NSG mice (n = 4) as a control.

For *in vivo* studies, SM4 (a kind gift from Gertrude Biomedical Pty Ltd., Australia) was freshly prepared in the vehicle solution of 80% Kollisolv PEG-400 (#06855; Sigma), 10% MilliQ water and 10% Kolliphor ELP/Solutol HS-15 (#42966; Sigma) for each treatment dosing day at a concentration of 8 mg/mL. One day after subcutaneous implanting of the cell-Matrigel suspension a dose of 25 mg/kg of body weight was administered to mice daily for 10 sequential days as described in [27] using disposable polypropylene 20 ga x 38 mm feeding tubes (#FTP-20-38; Instech, Philadelphia, USA) optimal for safe intragastric (IG) administration. The bioavailability of SM4 was not measured during the treatment in this study, as the drug has been shown, when using a similar dose and administration route of SM4, to be consistently detected in plasma of the treated mice, indicating a good systemic delivery of the drug [27]. The gait, piloerection, type of breathing, alertness, skin tone, eye condition and abdomen were all normal on handling, and no signs of dehydration, diarrhea, or other adverse reactions were observed during the treatment period. After the 10-day SM4 or Vehicle control treatment, and 24 h after the last dose, the mice were euthanized under anesthesia by cervical dislocation. In the autopsy, no abnormal swelling, colorization, or internal bleeding was observed. The injected cells (appearing as visible/palpable Matrigel plugs) were quickly removed from each mouse for either histological sampling and embedded in 10% neutral buffered formalin solution (#HT501128; Sigma) or stored in -80C for DNA extraction performed by NucleoSpin

Tissue Mini Kit (Macherey-Nagel) according to manufacturer's instructions. The histological samples were carefully cleaned of the mouse fat and skin and enclosed to tissue cassettes (Leica, Germany) for dehydration and processing with an overnight program in Tissue-Tek VIP 5Jr. (Sakura, Japan). The next day, the processed samples were embedded in paraffin (Sakura Tissue-Tek) and solidified into blocks at +4˚C until cut. The blocks were cut with a Microtome (Leica) to sections and dried overnight on glass slides until stored at +4˚C.

### Immunohistochemistry

Deparaffination, rehydration and epitope revealing were done before staining as described in [38]. Sections were stained with Hematoxylin (#1092530; Merck Millipore) and Eosin Y (#HT110132; Sigma) and for immunofluorescence using anti-rabbit GFP (a gift from Jason Mercer, University of Birmingham, UK) or anti-rat LANA (Abcam) primary antibodies and secondary anti-rabbit Alexa Fluor 488 or anti-rat Alexa Fluor 647 antibodies (#A11034, #21247; Invitrogen). Nuclear staining was done by incubating the sections in Hoechst 33342 (Sigma).

### Imaging and analysis

Images from the experiments on 96-well plates were taken using an automated cell imaging system ImageXpress Pico, and the quantifications were performed with the CellProfiler software. H&E sections were imaged with automatic Panorammic250 slide scanner with 20x microscope through services from Genome Biology Unit (GBU, University of Helsinki, Finland). Immunofluorescence images of spheroids and the *in vivo* whole mount sections were taken with Zeiss Confocal LSM 780 microscope provided by Biomedicum Imaging Unit (BIU, University of Helsinki, Finland). The images of sections were taken as a Z-stack and tiling was chosen to image the whole section area. GFP was imaged with a 20x objective using laser Argon 488nm and LANA was imaged with a 63x objective using laser HeNe 633nm. The images were quantified for GFP by the integrated ZEN 3.5 analysis program (Zeiss, Germany) and a pipeline was generated for the images to measure the relative GFP intensity normalized to the section area and the comparable coverage of the GFP signal in the cells showing an elongated spindling phenotype. Images of similarly prepared and stained histological samples from the implanted, mock infected lymphatic ECFC plugs, which do not express GFP, were used to subtract the autofluorescence for quantification.

### Statistical analysis

Graphical presentations and statistical analysis were generated with GraphPad Prism Software v8.0 (Dotmatics, San Diego, USA). For statistical evaluation of the RT-qPCR data for relative KSHV genome copies, the logarithmic values were converted to linear log2 scale values by using the double delta CT (2-ΔΔ CT) method. Human ALU-sequences were used as internal control and accounted in the calculations to correct differences in the DNA amount, quality, and PCR synthesis efficacy between the samples. The data is presented as individual values ± standard deviation (SD) between three biological replicates unless otherwise reported. Statistical differences between groups were evaluated with Student's *t*-test (two-tailed). Mean ± SD was shown and a p-value of ≤0.05 was considered significant and indicated by asterisk.

### Supporting information

**S1 Fig. Subpopulation of the ECFC isolates express markers of lymphatic vasculature.** Comparison of the adherent ECFC isolates to the mature LECs and BECs. **A.** Phase contrast

microscopic images. **B.** Expression of surface (CD31, CD34, VEGFR3, and Podoplanin) and nuclear (SOX18 and PROX1) endothelial cell markers analyzed by FACS. Corresponding IgG isotype antibodies were used as controls. **C.** The endothelial cell markers in B analysed by FACS from ECFCs isolated from four different healthy donors.
(TIF)

**S2 Fig. Comparison of infected lymphatic ECFCs from different donors.** ECFCLYs, isolated from four different donors, were infected with rKSHV.219. **A.** Pictures taken at 7 d.p.i show spindling phenotype, latent infection (GFP) and spontaneous lytic replication (indicated by RFP expression). **B.** Expression of KSHV latent (LANA) and lytic (K8.1) proteins and SOX18. **C.** KSHV titers were measured from virus release assay on naïve U2OS cells.
(TIF)

**S3 Fig. Gene expression changes induced by KSHV infection of lymphatic ECFCs. A.** Venn diagram showing overlapping gene expression profiles of wtKSHV-infected BEC and LEC along with blood and lymphatic ECFCs. **B** and **C.** Blood (black bars) and lymphatic (white bars) ECFCs were mock- or KSHV-infected. At 48 h.p.i, RNA was isolated and analyzed for gene expression of **(B)** a selection of genes and of **(C)** immune response genes identified as changed by RNA-sequencing. **D.** Volcano plot of K-ECFCBL and K-ECFCLY reads aligned to the KSHV genome indicating only one gene, K15, differentially expressed between the two cell types. **E-G.** Lymphatic ECFCs were mock or rKSHV.219-infected, and 7 d.p.i RNA was isolated and analyzed for gene expression by RNA-sequencing. Common gene expression changes between different isolates of ECFCLY after KSHV-infection are shown as Venn diagram **(E)**. Volcano blot with at least 2-fold up- (orange) and downregulated (black) genes with adjusted P value < 0.05 **(F),** and validation of selection of genes by qPCR **(G).** RNA-sequencing was done with biological triplicates of each sample.
(TIF)

**S4 Fig. SM4 inhibits the hallmarks of KSHV infection in the mixture of lymphatic ECFCs and LECs *in vitro*.** Lymphatic ECFCs and LECs were mixed at a ratio of 95% to 5% and infected with KSHV for five days. **A.** Cells were treated with 25 μM SM4 or DMSO control for six days, replenished at day three. The effect of the treatments is shown as quantification of KSHV genome copies from total DNA **(B)**, percentage of LANA positive cells on 96-well plates **(C)** and KSHV titers measured by the virus release assay on naïve U2OS cells **(D)**. **E.** Infected ECFC/LEC mixture was allowed to form spheroids overnight and then embedded in 3D fibrin. Spheroids were treated for six days with either 25 μM SM4 or DMSO control, fixed and imaged. Phase contrast and confocal images are shown at treatment day six. * p < 0.05, ** p < 0.01.
(TIF)

**S5 Fig. Histological sections from the mice engrafted with uninfected ECFCLYs.** Uninfected ECFCLYs were implanted subcutaneously into NSG mice and collected 30 days later **(A)** or after a 10-day treatment with either Vehicle or SM4 (25 mg/kg) **(B)** for histological analyses by IHC and IF.
(TIF)

**S1 Text. Gene Ontology categories and genes enriched in KSHV-infected blood ECFCs.**
(DOCX)

**S2 Text. Gene Ontology categories and genes enriched in KSHV-infected lymphatic ECFCs.**
(DOCX)

**S3 Text. Enriched gene sets in KSHV-infected LEC and ECFCLY cell types.**
(DOCX)

**S1 Table. Differentially expressed genes between KSHV-infected blood and lymphatic ECFCs.**
(XLSX)

**S2 Table. Shared differentially expressed genes between KSHV-infected lymphatic ECFCs 48h and 7d post infection.**
(XLSX)

## Acknowledgments

We thank the Laboratory Animal Center (LAC) HiLIFE, Biomedicum Imaging Unit (BIU), and Genome Biology Unit (GBU) at the University of Helsinki for support in animal care and imaging. We are also extremely grateful to Nadezhda Zinovkina, Hector Monzo, Shadi Azam and Vadim Le Joncour (University of Helsinki) for the valuable technical help. Mathias Fran-coís (The Centenary Institute, University of Sydney, Australia) is acknowledged for valuable comments for the manuscript. Fig 6A was created with BioRender.com.

## Author Contributions

**Conceptualization:** Krista Tuohinto, Terri A. DiMaio, Tara Karnezis, Michael Lagunoff, Päivi M. Ojala.

**Data curation:** Päivi M. Ojala.

**Formal analysis:** Krista Tuohinto, Terri A. DiMaio, Michael Lagunoff, Päivi M. Ojala.

**Funding acquisition:** Krista Tuohinto, Tara Karnezis, Michael Lagunoff, Päivi M. Ojala.

**Investigation:** Krista Tuohinto, Terri A. DiMaio.

**Methodology:** Krista Tuohinto, Terri A. DiMaio, Elina A. Kiss, Pirjo Laakkonen, Pipsa Saharinen.

**Project administration:** Päivi M. Ojala.

**Resources:** Tara Karnezis, Päivi M. Ojala.

**Supervision:** Michael Lagunoff, Päivi M. Ojala.

**Validation:** Päivi M. Ojala.

**Visualization:** Krista Tuohinto, Terri A. DiMaio.

**Writing – original draft:** Krista Tuohinto, Terri A. DiMaio, Elina A. Kiss, Pirjo Laakkonen, Pipsa Saharinen, Tara Karnezis, Michael Lagunoff, Päivi M. Ojala.

**Writing – review & editing:** Krista Tuohinto, Terri A. DiMaio, Michael Lagunoff, Päivi M. Ojala.

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
