## [Decision Letter · Decision Letter 0]

26 Aug 2022

Dear Prof Ojala,

Thank you very much for submitting your manuscript "KSHV infection of endothelial precursor cells with lymphatic characteristics as a novel model for translational Kaposi’s sarcoma studies" for consideration at PLOS Pathogens. As with all papers reviewed by the journal, your manuscript was reviewed by members of the editorial board and by several independent reviewers. In light of the reviews (below this email), we would like to invite the resubmission of a significantly-revised version that takes into account the reviewers' comments.

We cannot make any decision about publication until we have seen the revised manuscript and your response to the reviewers' comments. Your revised manuscript is also likely to be sent to reviewers for further evaluation.

Sincerely,

Dirk P. Dittmer, Ph.D.

Associate Editor

PLOS Pathogens

Shou-Jiang Gao

Section Editor

PLOS Pathogens

Kasturi Haldar

Editor-in-Chief

PLOS Pathogens

orcid.org/0000-0001-5065-158X

Michael Malim

Editor-in-Chief

PLOS Pathogens

orcid.org/0000-0002-7699-2064

Reviewer's Responses to Questions

**Part I - Summary**

Reviewer #1: Kaposi Sarcoma is a malignancy characterized by hyper-angiogenesis and highly proliferative spindle-shaped cells infected with Kaposi Sarcoma Herpesvirus (KSHV). The origin of these spindle cells has been elusive. Candidates include blood endothelial cells, lymphatic endothelial cells and mesenchymal stem cells. Previous work by this group indicated the presence of a circulating endothelial cell that harbored KSHV. Here the authors examine if circulating endothelial cell precursors termed endothelial colony forming cells (ECFCs) are permissive for KSHV infection and recapitulate aspects of KS. Major findings of their descriptive report are that: (1) lymphatic ECFCs were more permissive to KSHV and maintain viral episomes better than blood ECFCs; (2) infection of KSHV resulted in growth inhibition under 2D culture but increased survival in 3D culture; (3) and that infection has differential impacts on the host gene expression profile in lymphatic and blood ECFCs. They also confirm previous findings that Sox18 is a key host factor for lymphatic ECFCs to support KSHV infection. In summary, this well-written manuscript reports an exciting new primary cell type that is susceptible to infection and that may inform reservoirs of KSHV infection and pathogenesis in individuals. However, their characterization is superficial and it is unclear how these precursor cells compare to LECs and infected ECs in skin KS tumors.

Reviewer #2: This manuscript describes studies designed to identify the origin of the KS spindle cell, thus addressing an extremely important but unresolved question. The focus is on lymphatic endothelial cells (LEC), and more specifically on circulating endothelial colony forming cells (ECFCs) that express markers of lymphatic, not blood, endothelium. The premise that KSHV-infected lymphatic ECFCs are indeed the spindle cell precursors is based on a sound body of previous work contributed by the laboratories of the two corresponding authors, and by several independent studies. Additional benefits of this work include a comparison of KSHV infection dynamics in blood ECFCs and a demonstration that lymphatic ECFCs can be used in a mouse model for in vivo KS/KSHV translational studies.

Overall, this is a well-performed and accurately-reported body of work. However, a few additional experiments as well as minor clarifications and details are requested. For example, since an important component of this work’s contribution is method development, certain methods and materials should be described in more detail. In addition, the rationale for certain comparisons and the significance of the reported outcomes is not always clearly or sufficiently articulated. Finally, while I acknowledge the authors’ recognition that this work is in part intended to validate further characterization of the ECFC model in the future, there are a few additional experiments that would really benefit this particular study, in particular more work with sprouting assays and an in vitro validation of the cell mixing protocols used in the in vivo model.

Reviewer #3: Tuohinto, et al Review

KS spindle cells are known to be of endothelial origin, but it is not certain whether blood endothelial cells (BECs) or lymphatic endothelial cells (LECs) are the source. Another possibility is that KSHV infects circulating endothelial precursors known as endothelial colony forming cells (ECFCs). ECFCs themselves can be divided into blood and lymphatic subtypes. In this work the authors sought to better clarify which cell type ultimately forms spindle cells in KS tumors.

They found that lymphatic ECFCs, much like LECs, were more permissive to KSHV infection, maintained the viral episome longer, and produced more infectious virus than blood ECFCs or BECs. Lymphatic ECFCs also supported spontaneous lytic reactivation. In addition, it was determined that KSHV-infected lymphatic ECFCs, but not infected blood ECFCs or uninfected cells, were able to form small colonies in soft agar. Importantly, the authors were able to use lymphatic ECFCs to develop an in vivo model of KS and use it to test the efficacy of SM4 as a potential treatment.

This work not only addresses a significant question in the KSHV field, but provides a promising in vivo model which can be used to assess future drug candidates for the treatment of KS.

**Part II – Major Issues: Key Experiments Required for Acceptance**

Reviewer #1: 1. The status of KSHV infection was inferred nearly exclusively by reporter fluorescent proteins that do not reflect viral gene expression programs. Staining of LANA as a marker of latency and K8.1 (or other lytic protein) as lytic makers are suggested to give more clarity regarding the % of cells in either program. While host DEGS are defined based on RNAseq data, there is no comprehensive examination of viral gene expression in the K-ECFCLYs. It is difficult to understand the kinetics and lifecycle of KSHV in these cells without a better analysis.

2. In Figure 1, more controls are needed to better compare different cell types. Pictures taken with KSHV infected ECFCBL and uninfected cells are missing in Figure 1C, in Figure 1E and Figure 1G, the counterpart with K-ECFCBL is also missing. In Figure 1D, the WB for indicated proteins in ECFCBL and K-ECFCBL would be useful.

3. In Figure 2, how would the authors explain infection of KSHV slowed down infected cell growth at early time point (60hpi) in Figure 2A but had little effect when examined at late time point (5dpi) in Figure 2B? In Figure 2C, it seems that KSHV might be inducing autocrine growth factors. Did the authors examine for secreted factors that drive survival in the conditioned media of the growth factor reduced conditions. Would different time points post-infection be a factor in determining the outcome of the tube formation assay in 2D?

4. Figure 3A, while there were colonies observed with K-ECFCLY on day 36, the total cell number in the field was even lower than that with control ECFCLY. How do the authors distinguish colony formation from mere clustering? Could the authors better elaborate the difference between experiments in 3A and 3C where colonies seem improved? For the donors 1-4, cells per colony were not provided for the uninfected cells.

5. Supplemental Figure 3 would be better if incorporated into main Figures. Were there any DEGs that explain the nature of suppressed proliferation upon infection? Is SOX18 expression upregulated in ECFCLY upon KSHV infection? They include a venn diagram of LEC and ECFCLY with and without infection- does hierarchical analysis provide any insight regarding shifts in pathways caused by the virus that are common to both. Does KSHV infection drive the ECFCLY to a phenotype more like LECs?

6. In Figure 4, the authors did not report cell toxicity of SM4 at the doses tested. Figure 4 F-H studies the dose-depending effect of SM4 on KSHV replication in ECFCLY, while Figure 4E showed the effect of SM4 at 25um while Figure 4I and 4J reported the effect of SM4 at 50um. Cells in the spheroid of SM4 treated K-ECFCLY appear unhealthy.

7. It was quite curious that a mixture of ECFLY were implanted with LEC for the xenograft experiments. Are there any markers available to distinguish ECFCLY and LEC cells? Without a control implantation of K-ECFCLY alone, the tumorigenic potential of K-ECFCLY and the contribution of the LECs is unclear. As is stands now, this experiment does not inform in vivo tumor potential of the K-ECFCLY. The percentage of GFP varied dramatically from 20X to 63X, it is concerning that there is a high level of tissue autofluorescence in the 20X image that might influence ‘relative GFP intensity’. Was LANA used to confirm any of the GFP data for the xenograft analysis? SM4 at this concentration seems to be leading to tissue necrosis. Did the authors examine for markers of proliferation or cell death in the GFP+ infected cells compared to the uninfected cells in the tissue, with or without SM4?

Reviewer #2: 1. Figure 4I-J: The sprouting assays are described only in the context of SM4 treatment, yet they seem to be significant more generally, with respect to how KSHV infection affects ECFCs. A comparison of blood versus lymphatic ECFCs (as was done for proliferation and angiogenesis) should be done.

2. Figure 4A-H: concentrations of up to 50 microM SM4 were used. Please confirm that the DMSO concentration in all drug preparations was at the same level as in the DMSO only control. Also, it is important to provide a confirmation that the inhibitor is working in a SOX18-specific fashion in this system, for example by testing inhibition of SOX18-DNA binding.

3. Figure 5: For the mouse model, ECFCs were mixed with LEC prior to implantation. To validate this method and the exact contribution of the admixed LEC, some in vitro work with mixed cultures should be performed. Also, please clarify the meaning of the term ‘aberrant’ in Fig 5A. In Fig 5B: the difference in GFP intensity is interesting. Is this seen in vitro infection too? I assume yes but please clarify. Are the GFP-bright cells thought to be LECs or a ECFCs?

Reviewer #3: None noted

**Part III – Minor Issues: Editorial and Data Presentation Modifications**

Reviewer #1: Minor issues: Fig1C and Fig 2D ‘ECFCLY’ headers are split between two lines.

Reviewer #2: 1. In the Author Summary, lines 48-50, KSHV-induced transformation of lymphatic ECFCs is described as ‘minimal’. This descriptor when used in an absolute sense is accurate, but since it is being used here as a comparative term, a more appropriate term should be found. As it stands, it sounds as if the infected lymphatic ECFCs have less transformative potential than the other conditions tested, when in fact the opposite is true. Perhaps ‘modest’ would be a better term to use. Alternately the second part of the sentence could be revised to make the result clearer. I stress this relatively minor point because the author summary is a part of the paper that may be read first, making the need for clarity important.

2. Methods and Materials: The descriptions of the isolation of ECFCs in Seattle versus Helsinki appear to be different. The distinctions/similarities between the procedures should be clarified, and any difference in the phenotype of the final cell products noted. In addition, it should be indicated in the methods which EC preps (US vs. Finland) were used for the different experiments. Also, please confirm (if known) that blood products were from KSHV-negative adults, or indicate otherwise.

3. Methods and Materials: Please specify the type of tissue culture surface and growth medium used for the isolation and culture of ECFCs from the Puget Sound Blood Bank samples.

4. Methods and Materials: Infections were performed with different MOIs of KSHV. How was the amount of virus/infectious virus in the stocks determined?

5. Figure 1C-G (and other relevant figures): Were the samples shown from the high LOI or low LOI infection protocol?

6. Please provide additional comments on the testing of sub-populations of ECFCs versus (I assume bulk) ECFCs. What was the rationale for using these different isolation methods and can any useful conclusions be drawn from this comparison (from either a biological or a method optimization standpoint)?

7. Regarding experiments performed under limited growth factor conditions (lines 191-196), was a comparison of blood vs. lymphatic ECFCs under limited growth factor conditions ever performed? If yes, what was the result; if not this should also be noted. Also, what is the significance of this result?

8. Figure 2D: To improve rigor, some quantitation should be applied to this set of experiments, including with different donor samples and infections (e.g., counting of branch points and lumens). Were the cord structures monitored for longer than 24 hours post-plating? If yes, when did they start to lose stability and were the kinetics similar?

9. For the gene expression analysis, since blood ECFCs are less KSHV-permissive than their lymphatic counterparts, how was the degree of infection equalized?

10. Discussion: please comment on the significance of the observation that KSHV infection decreases EC proliferation. For example, does this have biological significance or is it an in vitro anomaly?

11. Typographical/grammatical corrections requested:

Line 67: change ‘induce’ to ‘induces’.

Line 154: change ‘capable to support’ to ‘capable of supporting’

Line 415: change ‘EFCFLYs’ to ‘ECFCLYs’

Lines 753-756: figure legend is poorly written. Please revise.

Reviewer #3: Comments

1. For the experiments shown in Figure 1, which compare blood and lymphatic ECFCs with BECs and LECs, ECFCBLs and/or BECs are often omitted. While the overall conclusions from these experiments are still well-justified, a more consistent presentation would be preferable.

2. It might be helpful to readers to include at least parts A and E of Supplemental Figure 3 as one of the main figures. At present there is no visual representation of the RNA-seq results to directly accompany the text.

3. The specific MOI used for the various infections is not indicated in most cases.

4. In the “siRNA Transfections” section of the Materials and Methods, it is not clear whether the amounts given are for each well or for the whole plate.

5. It might be helpful to include catalog numbers for reagents, if possible.

6. Figure 1 part C has a labeling error where the KSHV-infected ECFCLY group is just labeled “K-“.

7. There is a typo on line 274, where “blot” should be “plot”.

8. There is a typo on line 444, where “suing” should be “using”.

PLOS authors have the option to publish the peer review history of their article (what does this mean?). If published, this will include your full peer review and any attached files.

Reviewer #1: No

Reviewer #2: No

Reviewer #3: No
---

## [Decision Letter · Decision Letter 1]

2 Jan 2023

Dear Prof Ojala,

Thank you very much for submitting your manuscript "KSHV infection of endothelial precursor cells with lymphatic characteristics as a novel model for translational Kaposi’s sarcoma studies" for consideration at PLOS Pathogens. As with all papers reviewed by the journal, your manuscript was reviewed by members of the editorial board and by several independent reviewers. The reviewers appreciated the attention to an important topic. Based on the reviews, we are likely to accept this manuscript for publication, providing that you modify the manuscript according to the review recommendations.

Sincerely,

Dirk P. Dittmer, Ph.D.

Academic Editor

PLOS Pathogens

Shou-Jiang Gao

Section Editor

PLOS Pathogens

Kasturi Haldar

Editor-in-Chief

PLOS Pathogens

orcid.org/0000-0001-5065-158X

Michael Malim

Editor-in-Chief

PLOS Pathogens

orcid.org/0000-0002-7699-2064

Reviewer Comments (if any, and for reference):

Reviewer's Responses to Questions

**Part I - Summary**

Reviewer #1: The authors were very responsive to my concerns with modifications to all figures and their respective text descriptions, additional details for materials and methods, and the addition of more bioinformatic data. These changes deepen and strengthen the initial characterization of a novel lymphatic endothelial cell system in the field that supports KSHV infection and viral-driven colony formation and angiogenesis in cell culture and in immune deficient mice. This system is useful for investigating fundamental aspects of pathogenesis and for the investigation of therapeutics such as the SOX18 inhibitor SM4.

Reviewer #2: This manuscript describes the nature of KSHV infection in circulating endothelial colony forming cells (ECFCs) that express markers of lymphatic versus blood endothelium. Similar to primary LEC, lymphatic ECFCs, are more permissive to KSHV infection than blood ECFCs, support a degree of spontaneous lytic reactivation and form small colonies in soft agar. Importantly, lymphatic ECFCs can be used in a mouse model for in vivo KS/KSHV translational studies and here the utility of the SOX18 inhibitor SM4 is shown. This is a revised manuscript that has been significantly improved by additional method detail, new data and discussion comments. A few minor details still require attention.

**Part II – Major Issues: Key Experiments Required for Acceptance**

Reviewer #1: none

Reviewer #2: Regarding the comparison of KSHV genes in K-ECFCBL and K-ECFCLY in cells maintained under standard in vitro culture conditions, the relative gene expression patterns between the two cell types were remarkably similar, with the exception of K15 (higher in K-ECFCLY). This finding is shown in supplementary figure 3D and mentioned in the associated text. The authors should revisit this finding in the discussion and comment further on its potential significance.

Figure 4 includes new data showing the effect of SM4 on K-ECFCLY and ECFCLY 3D-sprouting at 25uM as well as 50uM doses, and on K-ECFCBY and ECFCBY 3D-sprouting but only at the 25uM dose. Was a 50uM dose tested on the blood-origin cells and if so what was the result? For example, were spheroids unaffected at the higher dose of SM4 or was non-specific toxicity observed, and did mock and KSHV-infected cells yield the same phenotype. Even if actual figure panels are not added, a comment should be added to the text.

**Part III – Minor Issues: Editorial and Data Presentation Modifications**

Reviewer #1: Minor issues remain regarding Figure 4. Panels A-I were not available in the Figure 4 download but there were no major issues in the original submission. With regard to Fig 4J-K, the bright field images do not match the IFA. Please clarify if these images were taken separately due to technical constraints.

Reviewer #2: The wording in the first few sentences of the Results section discussing the permissivity of the different cell populations to KSHV infection is a bit clumsy. One suggestion is to reword the third sentence (lines 126-127) as follows: “To determine whether these lymphatic ECFCs (ECFCLYs) resemble neonatal LEC with respect to an enhanced susceptibly to KSHV-infection, we seeded both ECFCLYs and blood ECFCs (ECFCBLs) onto 6-well….”

The comment on BEC not supporting lytic infection (line 143) sounds a bit dogmatic. I suggest modifying this statement so it is less definitive, for example: “Next, we analyzed if either population of ECFCs could support spontaneous lytic replication and production of new infectious viruses, a phenotype typically observed only in KSHV-infected LECs, but not BECs”

For the text in line 185, please replace “ended up with” with another descriptor, e.g., “harbored” or “maintained”.

Line 252: please change ‘to soft agar’ to ‘in soft agar’.

Lines 314-316: please change to ‘at the protein level’ and ‘the mRNA level’.

Line 330: please change to ‘concentrations of….’.

Line 368: please change to ‘a spindling phenotype…”

PLOS authors have the option to publish the peer review history of their article (what does this mean?). If published, this will include your full peer review and any attached files.

Reviewer #1: No

Reviewer #2: No

Figure Files:

Data Requirements:

Reproducibility:

References:

---

## [Editor Report · Decision Letter 2]

11 Jan 2023

Dear Prof Ojala,

We are pleased to inform you that your manuscript 'KSHV infection of endothelial precursor cells with lymphatic characteristics as a novel model for translational Kaposi’s sarcoma studies' has been provisionally accepted for publication in PLOS Pathogens.

Best regards,

Dirk P. Dittmer, Ph.D.

Academic Editor

PLOS Pathogens

Shou-Jiang Gao

Section Editor

PLOS Pathogens

Kasturi Haldar

Editor-in-Chief

PLOS Pathogens

orcid.org/0000-0001-5065-158X

Michael Malim

Editor-in-Chief

PLOS Pathogens

orcid.org/0000-0002-7699-2064
---

## [Editor Report · Acceptance letter]

20 Jan 2023

Dear Prof Ojala,

We are delighted to inform you that your manuscript, "KSHV infection of endothelial precursor cells with lymphatic characteristics as a novel model for translational Kaposi’s sarcoma studies," has been formally accepted for publication in PLOS Pathogens.

Best regards,

Kasturi Haldar

Editor-in-Chief

PLOS Pathogens

orcid.org/0000-0001-5065-158X

Michael Malim

Editor-in-Chief

PLOS Pathogens

orcid.org/0000-0002-7699-2064